# Geometric constraint-triggered collagen expression mediates bacterial-host adhesion

Yuting Feng[1], Shuyi Wang[1], Xiaoye Liu[2], Yiming Han[1], Hongwei Xu[1], Xiaocen Duan[1], Wenyue Xie[1], Zhuoling Tian[1,3], Zuoying Yuan[1], Zhuo Wan[1], Liang Xu[1,3], Siying Qin[4], Kangmin He[5,6] & Jianyong Huang [1] ✉

Cells living in geometrically confined microenvironments are ubiquitous in various physiological processes, e.g., wound closure. However, it remains unclear whether and how spatially geometric constraints on host cells regulate bacteria-host interactions. Here, we reveal that interactions between bacteria and spatially constrained cell monolayers exhibit strong spatial heterogeneity, and that bacteria tend to adhere to these cells near the outer edges of confined monolayers. The bacterial adhesion force near the edges of the micropatterned monolayers is up to 75 nN, which is ~3 times higher than that at the centers, depending on the underlying substrate rigidities. Single-cell RNA sequencing experiments indicate that spatially heterogeneous expression of collagen IV with significant edge effects is responsible for the location-dependent bacterial adhesion. Finally, we show that collagen IV inhibitors can potentially be utilized as adjuvants to reduce bacterial adhesion and thus markedly enhance the efficacy of antibiotics, as demonstrated in animal experiments.

Physical contact and subsequent adhesion between bacteria and host cells are prerequisites for bacterial infections[1,2], which have become one of the most serious threats to human health worldwide due to the presence of highly pathogenic bacteria and drug-resistant bacteria[3]. In the bacteria-host interactions, bacteria first reach the host surface with the aid of various physicochemical factors, likely achieving specific adhesion through adhesins and cellular receptors[4]. Some bacteria even internalize into the host cells in the subsequent process, which protects them from the host immune system and antibiotics[5–7]. There is growing evidence that physical microenvironment of bacteria and host cells, including but not limited to interfacial adhesion forces, extracellular matrix (ECM) rigidities, and geometric constraints[8–11], plays a key role in modulating their functions and behaviors and therefore affects various interactions between bacteria and host cells[12,13]. Bacterial adhesion force as an important factor in pathogenicity, for instance, can facilitate the transmission of bacterial toxins[14]. Recent advances in mechanobiology have already revealed that ECM rigidities

regulate bacteria–host interactions and bacterial internalization through cytoskeleton remodeling[15,16]. Also, geometric constraints on host cells, which are ubiquitous in various biological processes such as wound closure[17], cell migration[10,18], and tumor formation[19], are found to mediate cell morphology[18], cell–ECM adhesion[20], and cell–cell interactions[21]. It was reported that tissue morphogenesis and tumor progression are intrinsically regulated by the competition between cellular traction forces and intercellular contractile stresses[22]. Likewise, the bacteria–host interactions are implicated in complex biophysical interactions, where pathogens can manipulate mechanotransduction of host cells to facilitate dissemination while the host cells may alter their defense strategies to eliminate the pathogens[23,24]. For example, the virulence factor Sca4 secreted by *Rickettsia parkeri* specifically binds to vinculin to reduce intercellular tension, allowing bacteria to spread more easily by manipulating cytoskeletal forces[25]. In contrast, collective responses of epithelial monolayers to bacterial invasion can trigger the extrusion of infected

[1]Department of Mechanics and Engineering Science, College of Engineering, Peking University, 100871 Beijing, China. [2]Beijing Traditional Chinese Veterinary Engineering Center and Beijing Key Laboratory of Traditional Chinese Veterinary Medicine, Beijing University of Agriculture, 102206 Beijing, China. [3]Academy for Advanced Interdisciplinary Studies, Peking University, 100871 Beijing, China. [4]School of Life Sciences, Peking University, 100871 Beijing, China. [5]State Key Laboratory of Molecular Developmental Biology, Institute of Genetics and Developmental Biology, Chinese Academy of Sciences, 100101 Beijing, China. [6]University of Chinese Academy of Sciences, 100049 Beijing, China. ✉e-mail: jyhuang@pku.edu.cn

cells to limit local spread of infection[23,26]. Furthermore, gasdermin-D (GSDMD) pore-forming protein mediates the efflux of secretory vesicles from intestinal goblet cells in a nonpyroptotic manner to resist the invasion of pathogenic microorganisms, playing an important physiological role in intestinal barrier homeostasis[27]. Although there exist complicated mechanosensing and mechanotransduction in the bacteria–host interplay associated with surrounding microenvironments[17,28], whether and how spatially geometric constraints on host cells trigger their functional responses to regulate physical interactions with pathogenic bacteria still remain mysterious.

Here, we first develop an in vitro model of typical bacteria, e.g., *S. aureus* and *E. coli*, infecting epithelial cells, e.g., IEC-6 cells (rat small intestinal epithelial cell line-6) and HaCat cells (human keratinocyte cell line)[29], with the aid of the well-established microcontact printing technique, which allows pathogenic bacteria to infect host cell monolayers grown on spatially confined extracellular substrates. We demonstrate that there are remarkably spatially heterogeneous bacteria-host interactions owing to the presence of two-dimensional (2D) geometric constraints on host monolayers. With the single-cell force spectroscopy (SCFS) based on a fluidic force microscope (FluidFM)[30], we then quantify adhesion forces between bacteria and geometrically confined host monolayers and thus reveal that these forces are intrinsically spatially position and substrate stiffness dependent. Using single-cell RNA sequencing (scRNA-seq) and Monte Carlo (MC) simulations, we further clarify that heterogeneous expression of collagen IV in the host monolayers as a result of geometric constraint-triggered edge effects plays a pivotal role in modulating spatially heterogeneous bacteria-host adhesion. Finally, we show that collagen IV inhibitors are not only effective in reducing the heterogeneous adhesion, but also can act as antibiotic adjuvants to enhance antibiotic potency.

## Results

### Geometric constraints induced heterogeneous bacteria–cell adhesion

Using the polydimethylsiloxane (PDMS)-based micropattern printing technique[31], we confined monolayers of IEC-6 cells on collagen-coated polyacrylamide (PAAm) substrates with controlled geometric shape and size. As a control, cell monolayers with circular micropatterns (200 μm diameter) were infected with bacteria (*S. aureus and E. coli*) expressing green fluorescent protein (GFP) and FITC-labeled latex beads for 2 h, respectively. As shown in Fig. 1a, there were clear differences in the spatial distributions of bacteria and latex beads attached to the micropatterned host monolayers. The bacteria adhered primarily to the edge of the cell monolayers whereas the latex beads adhered uniformly throughout the micropatterned cell monolayers. In sharp contrast, we observed no significant difference in the spatial distribution of bacteria and beads on the cell monolayers without geometric constraints (Supplementary Fig. 1a). Also, there was no statistical difference in the spatial distribution of *S. aureus* on low-density IEC-6 cell patches with different densities/sizes (Supplementary Fig. 1b, c). The GFP/FITC intensities of the adherent bacteria and beads were further quantified to obtain spatially normalized fluorescence heatmaps. It indicated that the normalized intensities of adherent bacter*i*a at the edges (at a position with 100 μm radius) of the micropatterned cell monolayers were much higher than those at the centers, while the normalized intensity of beads was position independent (Fig. 1b).

To further assess the generality of our findings, we changed the geometric constraint parameters of micropatterned cell monolayers, including their size (100, 200, and 400 μm diameter) and shape (pentagon, triangle, cross, and ring-shaped). We demonstrated that the spatially heterogeneous adhesion phenomenon still remained unchanged, indicating that the bacteria–host adhesion behavior also emerged on geometrically constrained cell monolayers of different sizes and shapes (Fig. 1c and Supplementary Fig. 1d). Likewise, we observed that the bacteria–cell adhesion was related to extracellular

substrate rigidities, which were in the range of physiopathological rigidities of intestinal and skin tissues in vivo[32–34] (Fig. 1d–f, Supplementary Fig. 1e, f, and Supplementary Methods). The spatial heterogeneity of bacteria–cell adhesion was more pronounced as the underlying substrates stiffness increased (Fig. 1d–f). Subsequently, we explored whether the spatially heterogeneous adhesion was modulated by cell shape. To this end, we calculated cell area and circularity[35] (i.e., circularity = 4π × area/perimeter²) through segmenting the cell monolayers with the help of the well-developed Cellprofiler software. Statistical results indicated that there were no significant differences in the cell areas between the central and peripheral regions of the micropatterned cell monolayers cultured on the soft, medium, and stiff substrates (Supplementary Table 1). Likewise, cell circularity at the central and peripheral regions of the micropatterned cell monolayers was not significantly different from that in cell monolayers without geometric constraints (Supplementary Fig. 1g), regardless of soft, medium, or stiff PAAm substrates (Supplementary Fig. 1g).

Then, we investigated spatiotemporal distribution of bacteria infecting micropatterned cell monolayers. The fluorescence intensities of bacteria at the edges of micropatterned host monolayers showed an increasing tendency with the increase in the infection time from 1 to 2.5 h (Supplementary Fig. 2), which further confirmed the spatially heterogeneous adhesion phenomenon with 2D edge effects. Subsequently, we attempted to examine the effect of cell types and adhesion coating substances on PAAm substrates on bacteria–cell adhesion under geometric constraints. To this end, micropatterned cell monolayers formed by monoclonal IEC-6 and HaCat cells on collagen I adhesive islands were infected with *S. aureus*, respectively. Similar bacteria–cell adhesion phenomenon with 2D edge effects might be observed (Supplementary Fig. 3a–c), implying that it was less susceptible to host cell types adopted in experiments. Meanwhile, we noticed that spatially heterogeneous adhesion phenomenon persisted even when we replaced the adhesion coating substance from collagen I to polylysine, suggesting that it was also unaffected by specific adhesion coating substances (Supplementary Fig. 3d, e). These experimental findings showed that geometric constraints on host cell monolayers played an essential role in regulating the bacteria–host interactions.

### Bacteria–cell adhesion was correlated with their adhesion forces

To dissect physical interactions between bacteria and host cell monolayers under geometric constraints, we employed the FluidFM-based SCFS[14,30,36,37] to quantify the bacteria–cell adhesion forces. As shown in Fig. 2a, negative pressure from a pressure controller in the FluidFM was first utilized to tightly trap a single bacterium in the opening of the probe. Subsequently, the cantilever with the single bacterium approached a specific adherent cell at a preset speed, dwelt at a preselected force setpoint for a certain amount of time so that the bacterium might fully contact the underlying host cell, and finally retracted to complete the force spectrum experiment (Supplementary Fig. 4a and Supplementary Movie 1). The time-related retraction force was calculated via characterizing the deformation of the cantilever in the FluidFM. According to the classical Hooke's law, the adhesion force ($F_{ad}$) could be expressed as[38]

$$F_{ad} = USk \qquad (1)$$

where $U$ (unit: V) was the voltage difference at which the probe was subjected to an adhesion force to return to its initial state, $S$ was the cantilever sensitivity (unit: m/V) and $k$ denoted its spring constant (unit: N/m).

We quantitatively detected the physical adhesion interactions between *S. aureus* and micropatterned IEC-6 cell monolayers at different spatial positions, as exhibited in Fig. 2b, c. The average adhesion force at the edge position (-75 nN) was ~3 times higher than that at the center position (-25 nN; Fig. 2c, d and Supplementary Fig. 4b, c). Also, a

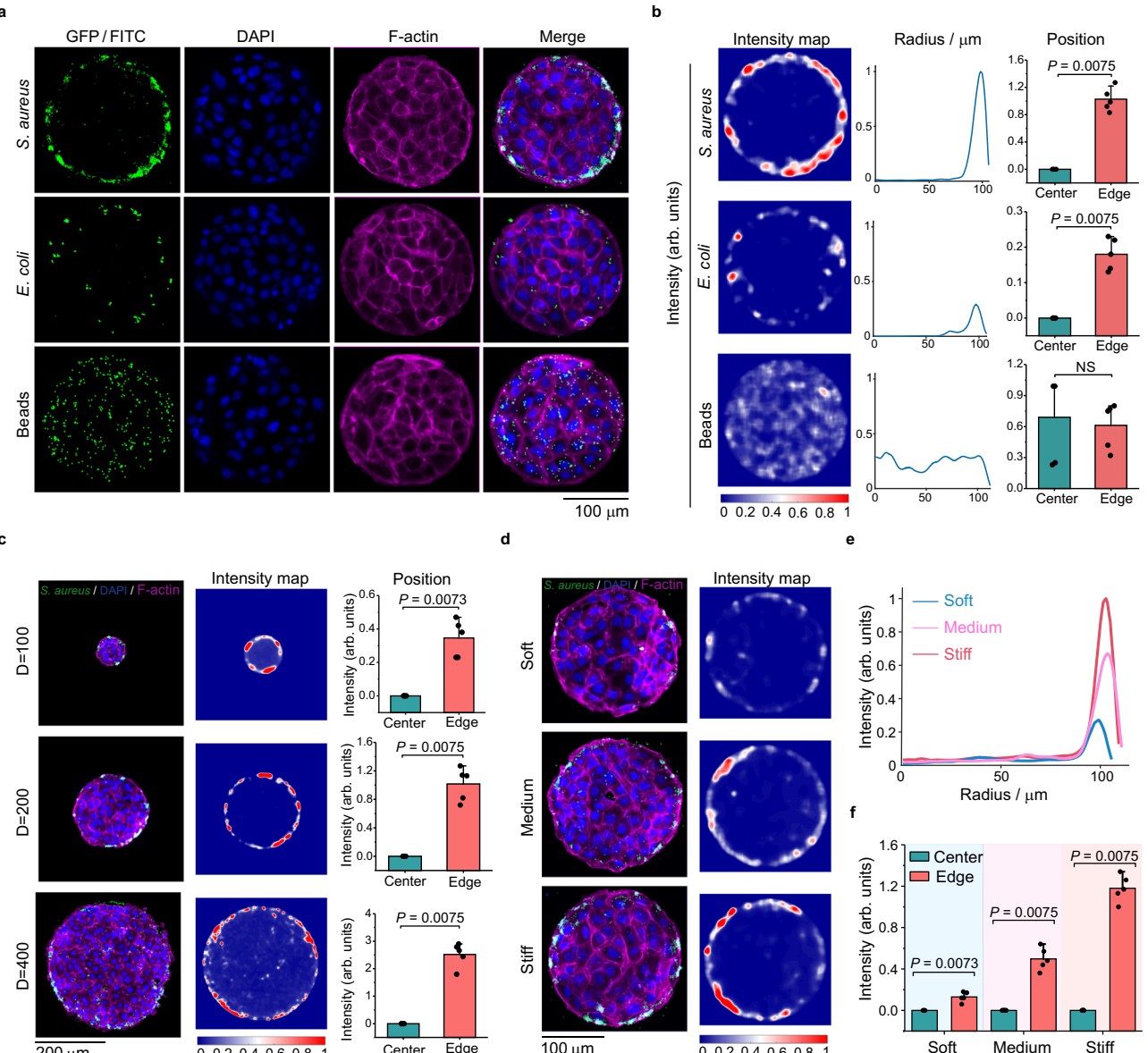

**Fig. 1 | Spatially geometric constraints on host cell monolayers triggered bacteria-cell heterogeneous adhesion. a** Representative images of circular monolayers of IEC-6 cells (200 µm diameter) infected by bacteria (*S. aureus and E. coli*) expressing GFP and yellow–green fluorescently labeled latex beads, respectively. For visualization, cell nuclei and F-actin cytoskeletons in the host cell monolayers were labeled with DAPI and Actin-Tracker Red-555 (Thermo Fisher), respectively. **b** Left: normalized fluorescence intensity distribution (heatmaps) of bacteria (*S. aureus and E. coli*) or latex beads attached to the micropatterned cell monolayers (200 µm diameter). Middle: average normalized fluorescence intensities along the radial directions of the micropatterns. Right: statistical comparisons of average normalized fluorescence intensities at the center ($r = 0$) and those at the edge ($r = 100$ µm) of the micropatterned cell monolayers. **c** Spatially heterogeneous adhesion results between *S. aureus* and circular monolayers of IEC-6 cells with different micropattern diameters (100, 200, and 400 µm). The intensity maps presented spatial distributions of normalized GFP intensities of adherent *S. aureus*. These histograms quantitatively compared the normalized GFP intensities at the center and edge of these circular cell monolayers involved. **d–f** Representative images of bacteria-cell heterogeneous adhesion interactions regulated by substrate rigidities, spatial distribution of normalized GFP intensities of adherent *S. aureus* (**d**), average normalized fluorescence intensities along the radial directions of the micropatterns (**e**), and statistical comparison (**f**) of the normalized GFP intensities at the center and the edge of these circular cell monolayers cultured on soft (10.14 kPa), medium (32.29 kPa), and stiff (93.46 kPa) substrates. All these measured data were shown as mean ± SD ($n = 5$ micropatterns from 5 independent experiments; two-tailed Mann–Whitney test). All representative data were repeated at least three times with similar results. Source data are provided as a Source Data file.

similar situation was observed for the adhesion interactions between *E. coli* and micropatterned IEC6 cell monolayers, where their average force was ~45 nN at the edge position whereas that was ~15 nN at the center position (Fig. 2e and Supplementary Fig. 4b, c). However, the adhesion forces between FITC-labeled latex beads and micropatterned IEC-6 cell monolayers were position-independent in essence (Fig. 2f and Supplementary Fig. 4b, c). Further, it turned out that the average adhesion force ($F_{ad}$, nN) between *S. aureus* and circular cell monolayers

(200 µm diameter) was approximately a linear function of the spatial position with $F_{ad} = 0.36\ r + 21.5$, where $r$ (µm) denoted the distance from the center of micropatterned monolayers (Fig. 2g). Interestingly, as the diameter of the circular cell monolayers increased from 100 to 400 µm, the average adhesion forces at their respective center positions remained roughly constant while those at their edge positions showed a slight fluctuation (Fig. 2h). These experimental results revealed that spatial distribution of bacteria−cell adhesion forces was

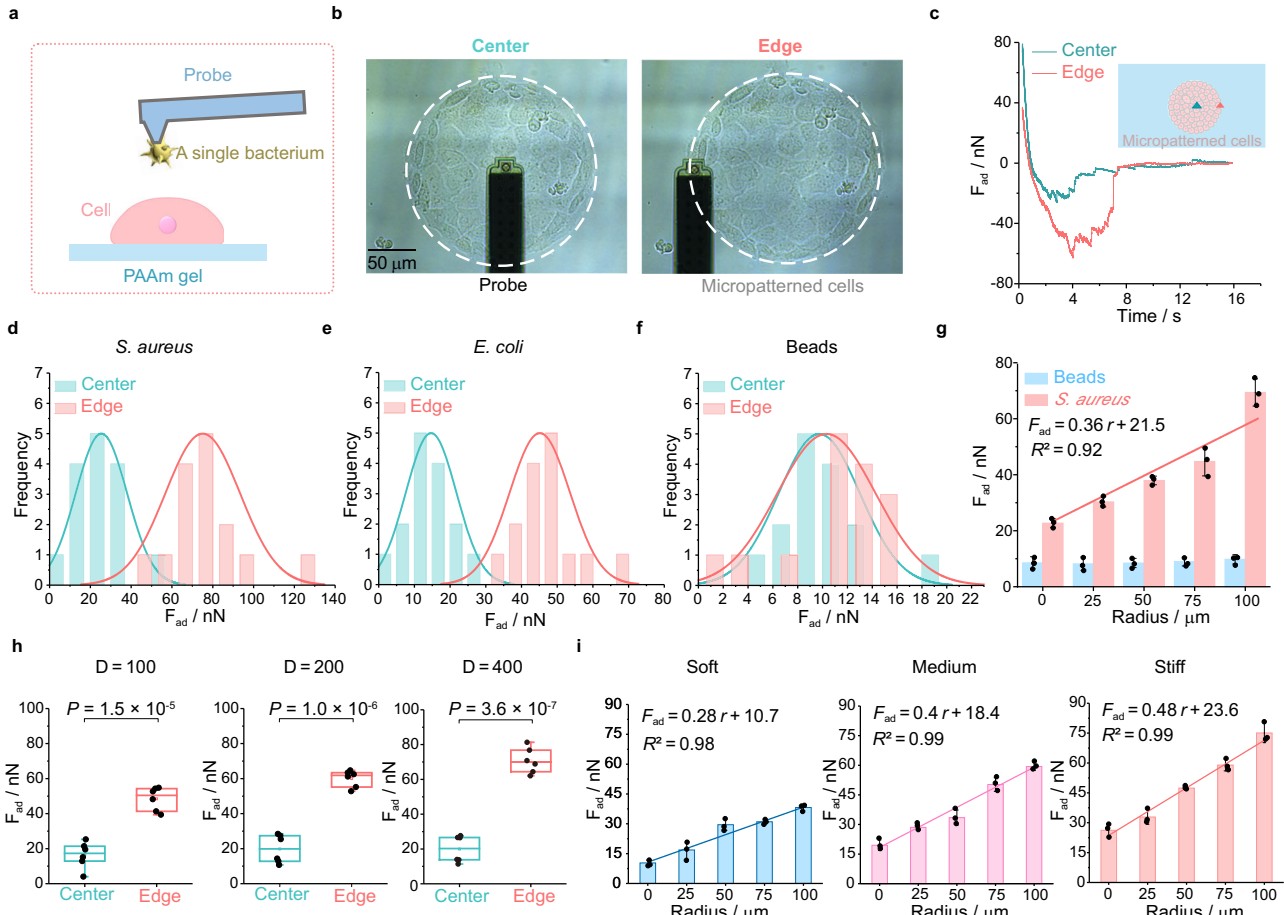

**Fig. 2 | Quantitative characterization of adhesion forces between single bacterium and micropatterned host cell monolayers with the FluidFM-based SCFS.** **a** Schematic diagram of quantifying the adhesion forces of a probe-adsorbed bacterium to cell monolayers grown on a PAAm hydrogel substrate with specific stiffness based on SCFS in a liquid microenvironment. **b** Representative images of bacterial adhesion force measurement at the center and at the edge of the micropatterned cell monolayers. The white dashed line represented the outer boundary of a circular micropatterned cell monolayer (200 μm diameter). Below the little ball at the tip of the black probe was the single bacterium trapped by negative pressure. **c** Typical time-related adhesion force ($F_{ad}$) curves between a single bacterium and host cells located near the center and edge of a micropatterned monolayer, which were obtained through the FluidFM-based SCFS. The adhesion force was calculated by quantifying the difference between the lowest point in a specific adhesion force curve and the corresponding one where the adhesion force finally reached a steady state with time. **d**–**f** Histograms of the measured adhesion forces of *S. aureus* (**d**), *E. coli* (**e**) and latex beads (**f**) to host cells located at the center and edge of these micropatterned cell monolayers cultured on stiff substrates, respectively ($n = 15$). The fitted curves showed spatial distributions of the measured data, whose peak positions denoted the average bacterial adhesion forces quantified by SCFS. **g** Comparison of adhesion forces for *S. aureus* and beads along the radial direction of the circular cell monolayers, which were measured by SCFS ($n = 3$). **h** Comparisons of adhesion forces between *S. aureus* and host cells located near the center and edge of these circular cell monolayers whose diameters were 100, 200, and 400 μm, respectively. Box plots indicate median (middle line), 25th, 75th percentile (box) ($n = 6$; two-tailed unpaired *t*-test). **i** Comparisons of bacterial (*S. aureus*) adhesion forces along the radial direction of the circular IEC-6 cell monolayers cultured on soft (10.14 kPa), medium (32.29 kPa) and stiff (93.46 kPa) substrates, respectively ($n = 3$). All representative data were repeated at least three times with similar results. Source data are provided as a Source Data file.

highly consistent with that of bacteria infecting cells; that is, the host cells with relatively high bacterial adhesion forces were more susceptible to bacterial infection, indicating that the bacteria–cell adhesion forces were regulated by geometric constraints on the host cell monolayers. We further investigated the effect of extracellular substrate stiffness on adhesion forces between bacteria and micropatterned cell monolayers. It appeared that the average adhesion forces were approximately linear with the distance from the center of the micropatterned monolayers on soft (10.14 kPa), medium (32.29 kPa), and stiff (93.46 kPa) substrates (i.e., soft: $F_{ad} = 0.28\,r + 10.7$; medium: $F_{ad} = 0.4\,r + 18.4$; and stiff: $F_{ad} = 0.48\,r + 23.6$) (Fig. 2i). The average adhesion forces at the edge positions on the stiff substrates were generally higher than those on medium and soft substrates, but the ratios of the average adhesion forces at the edge positions to the corresponding ones at the centers appeared to be independent of the diameters of the micropatterned cell monolayers and the underlying

substrate stiffness (Supplementary Fig. 4d, e). Additionally, we examined the effect of some key probe parameters, including probe speed and contact time between bacteria and host cells, on the measured results, which showed that the bacteria-cell adhesion forces gradually rose and finally reached a steady state with increasing the probe speed or contact time (Supplementary Fig. 4f, g), implying that the current experimental results were stable and reliable. Taken together, these findings indicated that spatial variations in the adhesion forces between bacteria and host cell monolayers were highly correlated with heterogeneous distributions of bacteria-cell adhesion induced by spatially geometric constraints.

## Geometric constraints triggered heterogeneous expression of collagen IV

To gain further insight into the molecular mechanisms underlying the differences in the spatial distribution of bacteria–cell adhesion forces

under geometric constraints, we performed scRNA-seq of IEC-6 cells located at the center and edge of the circular cell monolayers[39,40] (Fig. 3a). Array analysis quantitatively revealed striking differences in gene expression between cells at the center and edge of these host monolayers (Fig. 3b, c). Specifically, the gene *Col4a1* encoding collagen IV associated with bacterial adhesion (e.g., *S. aureus*, *E. coli*, *Streptococcus pneumoniae*, *Helicobacter pylori*, and *E. faecalis*)[41–45] was markedly increased in the cells near the edges of the circular cell monolayers, whereas the gene *Duox2* participating in host defense response to protect against infection[46,47] was significantly upregulated in those near the centers of the cell monolayers (Supplementary Fig. 5). These data indicated that geometric constraints on host cell monolayers indeed regulated their gene phenotypes and thus caused spatial differences in cellular gene expression.

Further, we detected the spatial distribution of collagen IV on the circular cell monolayers based on immunofluorescence staining (Fig. 3d), which showed that collagen IV expression was mainly distributed at the edges of these cell monolayers cultured on PAAm substrates with different rigidities. Interestingly, there was no spatial heterogeneity in the expression of other types of collagen proteins, such as collagen I and collagen II (Supplementary Fig. 6a–c), illustrating the specificity of collagen IV expression mediated by spatially geometric constraints. We also formed low-density IEC-6 cell monolayers without geometric constraints on the stiff substrates and then performed immunofluorescence staining for collagen IV in the specific cell monolayers (Supplementary Fig. 6d, e). It was found that there was no spatially heterogeneous collagen IV expression in the cell monolayers without geometric constraints, implying that geometric constraints imposed on the cell monolayers played an essential role in triggering collagen IV expression with edge effects. Moreover, the amount of collagen IV expressed near the edges of the cell monolayers was positively correlated with the underlying substrate rigidities (Fig. 3d and Supplementary Fig. 7a, b), implying that ECM/substrate stiffness could exert an important impact on regulating the production of collagen IV in the host cell monolayers as well. Next, we employed a collagen IV inhibitor[48], i.e., VU6015929, to treat the micropatterned host cell monolayers cultured on soft, medium, and stiff PAAm substrates, respectively, and then investigated the interactions of bacteria and the host cell monolayers. It was found that the expression of collagen IV on all these cell monolayers was effectively inhibited by VU6015929 and that the phenomenon of heterogeneous bacteria-cell adhesion disappeared (Fig. 3e–g and Supplementary Fig. 7c, d). At the same time, all these bacteria-cell adhesion forces were significantly reduced and there were no significantly spatial differences in the measured adhesion forces after these host cell monolayers were treated with VU6015929 in the experiments (Fig. 3h and Supplementary Fig. 7e, f).

To verify the contribution of bacterial collagen adhesin (Cna) which may bind specifically to collagen IV during bacteria-cell adhesion[49], we subsequently infected micropatterned cell monolayers cultured on the stiff substrates using the wild-type *S. aureus* USA300 (wild) and *S. aureus* USA300 with knockout Cna gene (△Cna), respectively (Supplementary Fig. 7g). One could find that the Cna-deficient *S. aureus* (△Cna) no longer had significant edge effects during its interactions with the micropatterned cell monolayers with spatially geometric constraints and that the adhesion forces between the Cna-deficient bacteria and the cells located at the edge areas were significantly reduced so that spatial differences in the bacteria-cell adhesion forces also disappeared (Supplementary Fig. 7g–i). These indicated that specific binding of the bacterial collagen adhesin to collagen IV might be a key factor leading to the formation of spatially heterogeneous bacteria-cell adhesion. With the Kyoto Encyclopedia of Genes and Genomes (KEGG) results from the scRNA-seq, we further checked the focal adhesion pathway and its associated genes (e.g., vinculin) that were normally closely related to cellular traction forces, ECM stiffness and even bacterial infection[15,26,50] (Supplementary

Fig. 5b). In combination with immunofluorescence staining for vinculin in the micropatterned cell monolayers, we confirmed that the vinculin expression was also increased near the edges of the micropatterned cell monolayers (Supplementary Fig. 8a). Likewise, our quantitative data revealed that cellular traction forces tended to increase with increasing the underlying substrate rigidities, and that the forces near the edges of the micropatterned cell monolayers were much larger than those near the central regions, which was roughly consistent with spatial distribution of the collagen IV expression in the cell monolayers under geometric constraints (Supplementary Fig. 8b, c and Supplementary Methods). However, the heterogeneous expression of collagen IV and the resulting bacteria–host adhesion phenomenon were abolished after the micropatterned cell monolayers were treated with cytochalasin D that was a potent inhibitor of actin polymerization (Supplementary Fig. 8d, e). Interestingly, the Young's moduli of cells near the edge regions were lower than those near the central regions in the micropatterned cell monolayers, which was somewhat correlated with the spatial distribution of collagen IV expression[51] (Supplementary Fig. 8f–h and Supplementary Methods). In summary, these experimental findings suggested that geometric constraints on host cell monolayers might trigger the expression of collagen IV proteins with pronounced edge effects, which was responsible for the spatially differential bacteria-cell adhesion forces and the heterogeneous adhesive interactions between bacteria and the cell monolayers.

## Monte Carlo simulations revealed heterogeneous adhesion dynamics

To demonstrate the inherent mechanism by which geometric constraints on host cell monolayers modulated bacteria–cell heterogeneous adhesion, we proposed a microscopic kinetic model based on MC simulations in combination with a mechanochemical coupling framework. For simplicity, we ignored the effects of bacterial reproduction and bacteria–bacteria interactions. We described random motion of a single bacterium under gravity in a liquid environment by a Langevin equation[52]

$$m\frac{d^2\mathbf{r}}{dt^2} = \mathbf{F}_D + \mathbf{F}_B + \mathbf{F}_G + \mathbf{F}_{buo} \quad (2)$$

where $m$ was the mass of the bacterium, $\mathbf{r}$ was its position vector, $t$ was the time, $\mathbf{F}_D$, $\mathbf{F}_B$, $\mathbf{F}_G$ and $\mathbf{F}_{buo}$ denoted the drag force, Brownian random force, gravity, and buoyancy, respectively (See Supplementary Information). Generally, there may be receptor-ligand interactions between the swimming bacterium with the underlying host cells monolayers. For instance, collagen IV receptors on host cell membranes can specifically bind to the Cna expressed by *S. aureus*, which is similar to the high-affinity "dock, lock and latch (DLL)" mechanism[49], as illustrated in Fig. 4a. The receptor-ligand interactions could be quantitatively characterized through a mechanochemical coupling model satisfying the following reversible random chemical process[53,54] (Fig. 4b and Supplementary Fig. 9a)

$$Cna + Col4 \underset{k_{a-}}{\overset{k_{a+}}{\rightleftarrows}} Bound \quad (3)$$

where $K_a$ indicated the reaction equilibrium constant satisfying the Boltzmann distribution[53], i.e.,

$$K_a = \frac{k_{a+}}{k_{a-}} = \exp\left(\frac{E_a}{k_B T}\right) \quad (4)$$

in which, $k_{a+}$ was the association rate, $k_{a-}$ was the dissociation rate, $E_a$ denoted the total activation energy of bacteria–cell adhesion, $k_B$ was the Boltzmann constant ($1.38 \times 10^{-23}$ J/K) and $T$ stood for the absolute temperature. The total activation energy required for the bacteria–cell

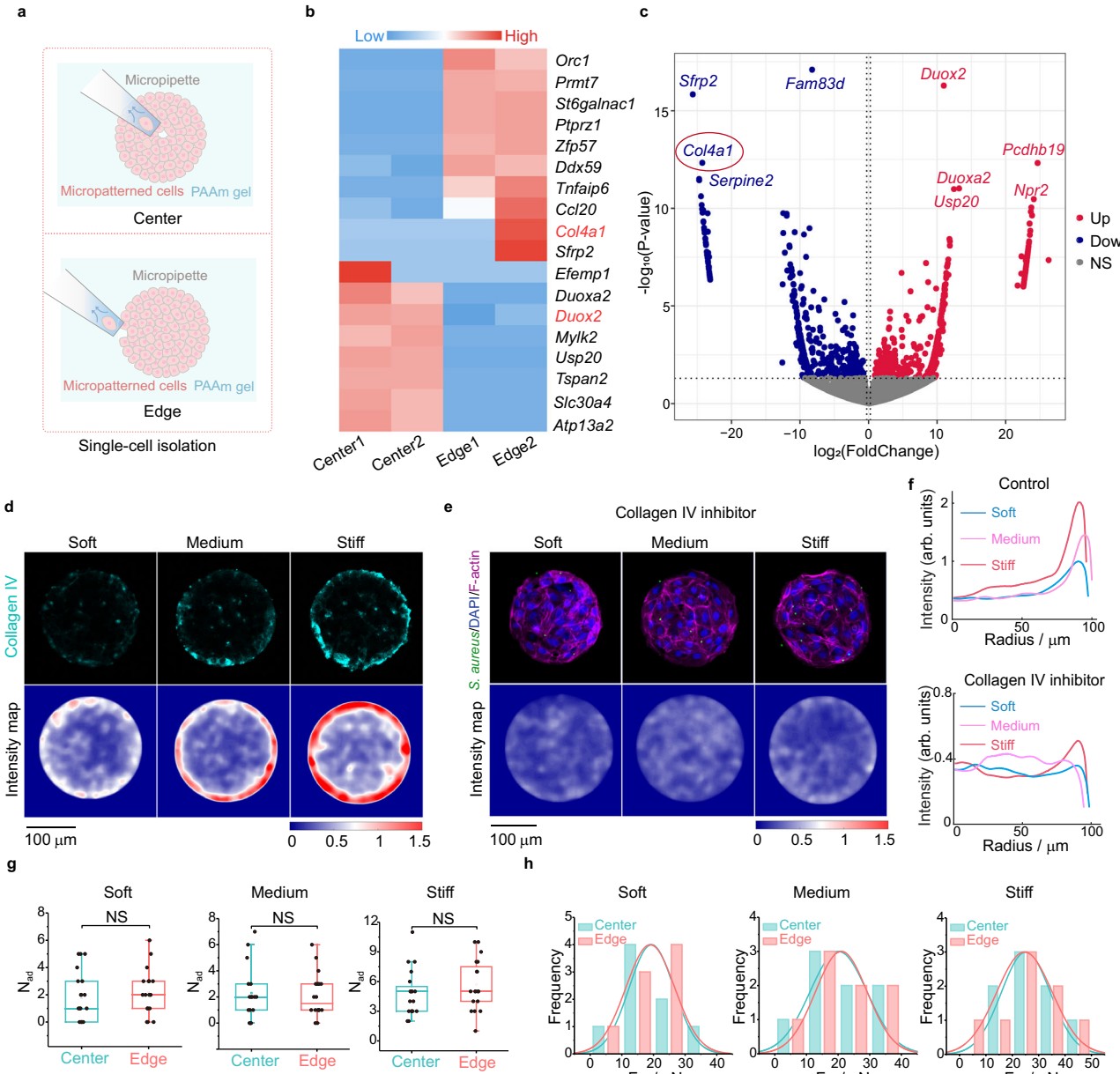

**Fig. 3 | Spatially heterogeneous expression of collagen IV on the micro-patterned host cell monolayers. a** Schematic diagram of single cell isolation and extraction by micropipette at the center and edge of an micropatterned IEC-6 cell monolayer (200 μm diameter). A mouth pipette (30 μm diameter) with negative pressure was adopted to sort the single cells into the lysate to extract RNA for single-cell RNA sequencing. **b** Heat map of differential gene expression levels, where red represented relatively high expression of genes involved, while blue denoted relatively low expression. **c** Volcano plot revealing differences in gene expression levels between host cells at the center and edge of micropatterned cell monolayers. Red, blue and gray dots represented genes that were upregulated (Up), down-regulated (Down) and not significantly (NS) different in the cells near the central regions relative to those near the edge regions of the micropatterned cell monolayers, respectively. **d** Immunofluorescence staining results (cyan) of collagen IV expressed in the cell monolayers cultured on soft, medium and stiff substrates, and the corresponding intensity maps. **e** Representative images of bacterial adhesion to host cell monolayers and intensity maps of collagen IV after inhibition of collagen IV. **f** Spatial changes in the mean fluorescence intensities of collagen IV along the radial direction of a circular cell monolayer before and after treatments with the collagen IV inhibitor. **g** Comparisons of the number of bacteria adhered ($N_{ad}$) near the center and edge of these circular host cell monolayers with different stiffness treated with the collagen IV inhibitor ($n = 20$; two-tailed unpaired $t$-test). Box plots indicate median (middle line), 25th, 75th percentile (box).
**h** Histograms and the corresponding fitted curves of the measured adhesion forces of bacteria to the cells located at the center and edge of these micropatterned cell monolayers treated with the collagen inhibitor, respectively ($n = 8$). All data were from at least three independent experiments. All representative data were repeated at least three times with similar results. Source data are provided as a Source Data file.

adhesion process could be represented by a simple linear spring model[53]

$$E_a = f\gamma = \frac{f^2}{2k} \qquad (5)$$

where $f$ denoted the adhesion force quantified by SCFS, $\gamma$ was the length scale parameter, and $k$ was the stiffness of these receptor-ligand bonds (See Supplementary Table 2 and Supplementary Methods for more details).

The MC simulations reproduced the spatiotemporal dynamics involved in random movement of single bacteria in a liquid

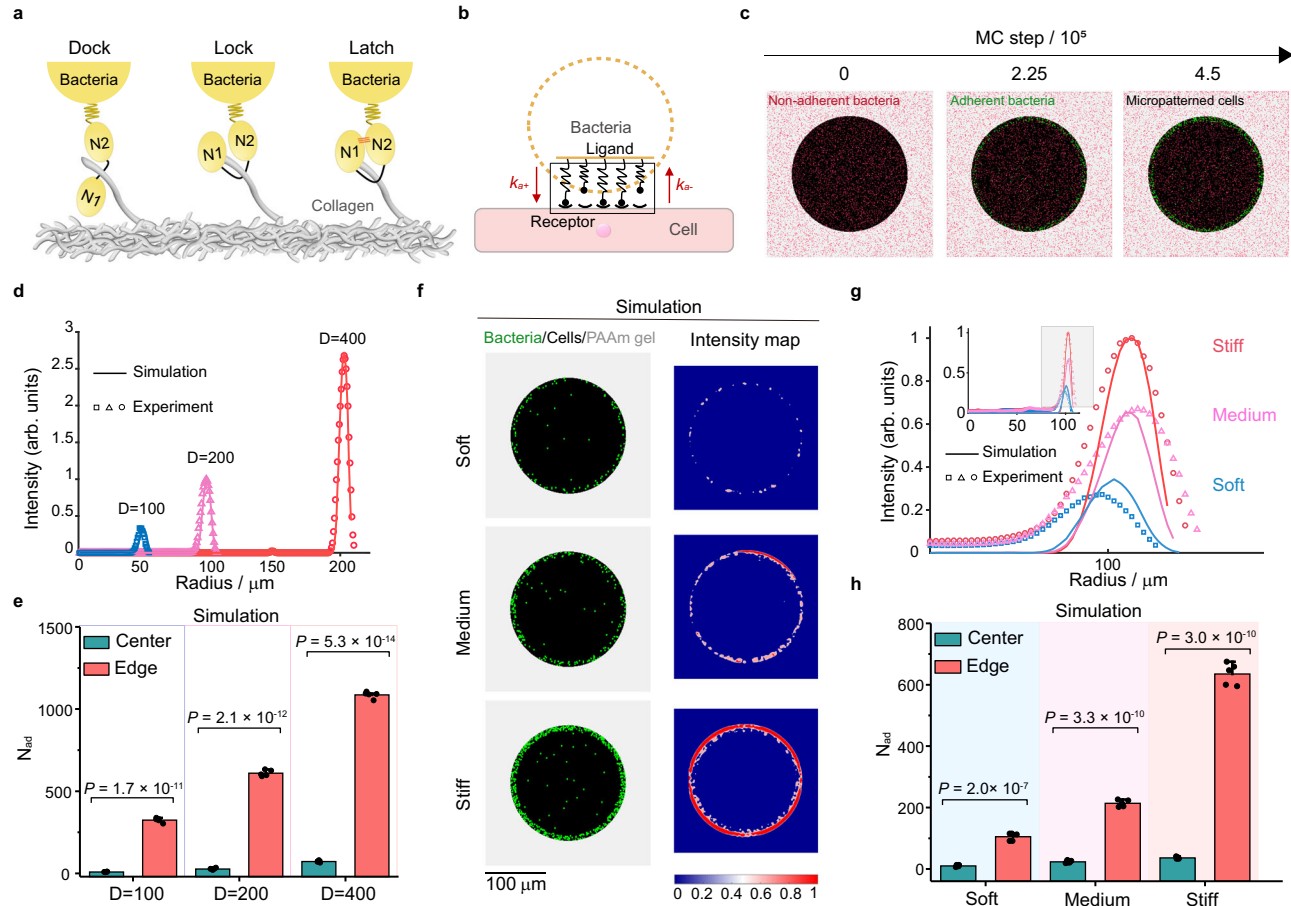

**Fig. 4 | Monte Carlo simulations of bacterial adhesion onto micropatterned host cell monolayers. a** Schematic diagram of DLL mechanism. N1 and N2 were two subdomains of Cna that wrapped collagen molecules (gray) through conformational changes. **b** Schematic diagram of bacteria–cell adhesion at the single-cell level. $k_{a+}$ denoted the association rate whereas $k_{a-}$ represented the dissociation rate. **c** Typical simulation results characterizing the bacteria–cell adhesion under geometric constrains at different Monte Carlo steps (MC step). Non-adherent bacteria were marked as red solid squares, and adherent bacteria were labeled as green solid squares. Micropatterned cells were denoted with a black background. At least three independent experiments were performed in each specific situation. **d** Comparison of average fluorescence intensities of adherent bacteria along the radial direction obtained from the simulations and the corresponding experiments. The diameter of the simulated host cell monolayers was set as 100, 200, and 400 μm, respectively. **e** Simulation data showing significant differences in the number of adherent bacteria ($N_{ad}$) onto the center and edge of the circular cell monolayers with different diameters (100, 200, and 400 μm). All the simulated data were displayed as mean ± SD ($n = 5$; two-tailed unpaired $t$-test). **f** Simulation results presenting spatial distributions of adherent bacteria onto the host cell monolayers cultured on soft (10.14 kPa), medium (32.29 kPa) and stiff (93.46 kPa) substrates, where the adherent bacteria were labeled as green solid squares, the micropatterned cell monolayers were marked with a black background, and the underlying PAAm hydrogel substrates were shown in gray. **g** Comparison of average fluorescence intensities of adherent bacteria along the radial direction obtained from the simulations and the corresponding experiments, in which the diameter of the circular cell monolayers was 200 μm. **h** Simulation data showing significant differences in the number of adherent bacteria onto the center and edge of the circular cell monolayers cultured on soft, medium and stiff substrates ($n = 5$; two-tailed unpaired $t$-test). All representative data were repeated at least three times with similar results. Source data are provided as a Source Data file.

environment, physical contact between bacteria and host cell monolayers, receptor-ligand binding and dissociation. The numerical results showed that, because of relatively high activation energy, bacteria were more likely to adhere to the regions near the edges of host cell monolayers, which was in good agreement with the experimental data (Fig. 4c and Supplementary Movie 2). By contrast, the latex beads tended to adhere uniformly onto the micropatterned host cell monolayers, as confirmed in the simulations (Supplementary Fig. 9b, c and Supplementary Movie 3). Also, the MC simulations could intuitively reflect the regulation of ECM/substrate stiffness and geometric parameters, e.g., diameter of the circular cell monolayers, on the bacteria-cell heterogeneous adhesion (Fig. 4d–h, Supplementary Fig. 9d, e, and Supplementary Movie 4). For example, a systematical comparison indicated that the normalized fluorescence intensities along the radial direction in the circular cell monolayers presented in the simulations were quantitatively consistent with those detected in

our experiments (Fig. 4d). Meanwhile, the simulation results presented that the heterogeneous distribution of adherent bacteria on the circular cell monolayers was substrate stiffness-dependent, which correlated well with the corresponding experimental data (Fig. 4g). All these simulation results further confirmed that the heterogeneous expression of collagen IV and the resultant spatially varying adhesion forces on the micropatterned cell monolayers were responsible for the spatially heterogeneous interactions between bacteria and host cell monolayers under geometric constraints.

## Collagen IV inhibitors synergistically potentiated antibiotic efficacy

Given that the collagen IV inhibitor, i.e., VU6015929, had been shown to be effective in reducing bacteria-cell adhesion in vitro, we subsequently investigated whether it had similar effects in vivo. To this end, we introduced a rat-excised wound infection model (wound of 1 cm

diameter) to systematically evaluate the efficacy of the collagen IV inhibitor as a potential antibiotic adjuvant against bacterial infection in vivo. In the wound infection model, the formed wounds were first treated with phosphate-buffered solution (PBS). Then, they were treated with *S. aureus* ($2.5 \times 10^8$ CFU/cm²) and PBS (PBS group), *S. aureus* and 1 μg/mL ciprofloxacin (0.1×MIC group), *S. aureus* and 10 μg/mL ciprofloxacin (1 × MIC group), *S. aureus* and 50 μM collagen IV inhibitor (VU6015929) (Inhibitor group), or *S. aureus*, 50 μM collagen IV inhibitor (VU6015929) and 1 μg/mL ciprofloxacin (Inhibitor + 0.1 × MIC group) on day 0, respectively (Fig. 5a).

Next, we quantified *S. aureus* colonies adhering to wounds using the colony-count technique on days 1 and 4, respectively. As shown in Fig. 5b, the number of *S. aureus* detected in the Inhibitor + 0.1 × MIC group was consistently lower than that in the other groups. These experimental findings indicated that the collagen IV inhibitor could be utilized as a potential antibiotic adjuvant against bacterial infection in vivo. Also, we performed Gram staining of wound sections on day 1 and day 4 to investigate the distribution and quantity of bacteria adhered to the wounds. It was found that bacteria were mainly attached near the wound edges on day 1, which was consistent with the results of in vitro experiments. However, we observed a significant reduction in the number of adherent bacteria in the wounds on day 4 (Fig. 5c). Besides, we investigated the effect of collagen IV inhibitors on wound healing and inflammation. The wound size and the expression of the inflammatory factor IL-6 in the Inhibitor + 0.1 × MIC group were smaller than those in the control groups (i.e., the PBS group and 1 × MIC group), which implied that the combination of inhibitors and antibiotic may accelerate wound closure and simultaneously reduce the inflammatory response to some extent (Fig. 5d, e and Supplementary Fig. 10). Normally, epidermal thickness increased during the initial inflammatory phase and decreased during the remodeling phase[55]. Our experimental results demonstrated that on day 10, the Inhibitor + 0.1 × MIC group had the thickest epidermis and shortest wound length, implying that the use of collagen IV inhibitor (VU6015929) could promote wound healing to a certain extent (Fig. 5f–h). All these findings suggested that the collagen IV inhibitor might effectively reduce bacterial adhesion onto host cells, synergistically enhance the efficacy of antibiotics and boost wound healing.

## Discussion

Previous reports demonstrated that cells in a monolayer could sense their surroundings and accordingly alter their phenotype or morphology through a cascade of signal transduction events to better survive when confined or crowded[56–59]. Particularly, cells located near the edges of some micropatterned cell monolayers could generate greater cytoskeletal contractility and traction stresses in response to spatially geometric constraints, which then mediated the expression of integrin α5β1 and molecular marker CD44 associated with cancer stem cells[60,61]. Similarly, our experimental results showed that epithelial cells (e.g., IEC-6 cells) in a micropatterned cell monolayer experienced spatially heterogeneous expression of collagen IV proteins, which hence caused spatially position-dependent bacteria-cell interactions. As quantified by the FluidFM-based SCFS, their average bacterial adhesion forces near the edge of these micropatterned cell monolayers were several times higher than those at the center, which also depended on ECM/substrate rigidities to some extent. These results not only uncovered the critical role of physical cues in host cell microenvironments including spatially geometric constraints and substrate rigidities, in the regulation of cellular phenotypes, but also reported the heterogeneous adhesion phenomenon between pathogenic bacteria and host cell monolayers.

As a matter of fact, the heterogeneous adhesion phenomenon with distinct edge effects was not limited to the interactions between *S. aureus* and IEC-6 endothelial cell monolayers. Similar phenomena could also be observed in experiments on the interactions between

other types of bacteria, e.g., *E. coli*, and host cell monolayers, e.g., HaCat cell monolayers (Fig. 1a and Supplementary Fig. 3a–c), suggesting that the geometric constraint-triggered heterogeneous adhesion was essentially less dependent on specific types of bacteria and host cells. Interestingly, some recent studies on bacterial invasion found that, unlike *Listeria monocytogenes* that was highly invasive to endothelial cells cultured on relatively stiff extracellular substrates[62], *S. aureus* exhibited relatively low bacterial invasion of IEC-6 cell monolayers grown on the stiff substrates[15] (Supplementary Fig. 11 and Supplementary Methods). This suggests that there are differences in bacterial invasion between different ECM rigidities, bacterial types and cell types. Furthermore, it should be pointed out that the adhesion forces between *S. aureus* and IEC-6 cell monolayers were higher than those between *E. coli* and the same cell monolayers (Fig. 2d, e), meaning that the absolute magnitude of bacterial adhesion forces might depend on types of bacteria involved. This was most likely due to the potential differences in the density of Cna ligands and physical adhesion energy on the surfaces of *S. aureus* and *E. coli*. Likewise, the high spatial correlation between the bacteria-cell adhesion forces and the expression of collagen IV on the micropatterned host cell monolayers prompted us to revisit bacterial infection and antibiotic treatment strategies from a mechanobiological perspective. Normally, physical contact and effective adhesion of bacteria to host cells were essential for bacterial infection. In this sense, effectively inhibiting the expression of specific adhesion receptors on the host cell surfaces might minimize the biomechanical adhesion between bacteria and host cells, thus facilitating the antibiotic treatment of bacteria. With the collagen IV inhibitor (VU6015929) as an antibiotic adjuvant in the animal experiments, one could find that, a 90% reduction in MIC of antibiotics could also achieve a comparable antimicrobial therapeutic effect (Fig. 5). This point was of vital importance for enhancing antibiotic potency and therefore avoid clinical misuse.

## Methods

### Ethics statement

All animal experiments described in this study were complied with the guidelines of Tianjin Medical Experimental Animal Care, and the animal protocols were approved by the Animal Ethics Committee of Yi Shengyuan Genome Technology (Tianjin) Co., Ltd. (protocol number YSY-DWLL-20211031).

### Fabrication of polyacrylamide substrates

Polyacrylamide (PAAm) gels of 10.14, 32.29, and 93.46 kPa were fabricated on the activated coverslip[63]. Briefly, the coverslip (20 mm diameter) was dried at room temperature (RT) after washing with methanol, activated with 3-aminopropyltrithoxysilane (APES, 4% v/v in acetone) until dry, and then washed with double distilled water (ddH2O) for 30 min. Finally, the activated coverslip was soaked in 0.5% glutaraldehyde for 30 min, washed three times with ddH2O and dried at RT. A solution containing acrylamide, bis-acrylamide (BIS), ammonium persulfate (APS), and tetramethylethylenediamine (TEMED) was prepared for PAAm gels with different stiffnesses (Supplementary Table 3). The mixed solution (~25 μl) was dropped onto the activated coverslip and quickly covered with another clean coverslip for 20 min at RT. The upper coverslip was removed carefully to obtain the PAAm substrate after waiting for the prepolymer solution to polymerize.

### Cell patterning on PAAm substrates

Polydimethylsiloxane (PDMS) stencils with recessed micropatterns containing circles of different diameters (100 μm, 200 μm, and 400 μm) and other shapes (pentagon, triangle, cross, ring-shaped) were prepared using standard soft lithography as explained previously[31]. The agarose stamps with raised micropatterns were transferred from PDMS stencils and then covered with 0.2 mg/mL

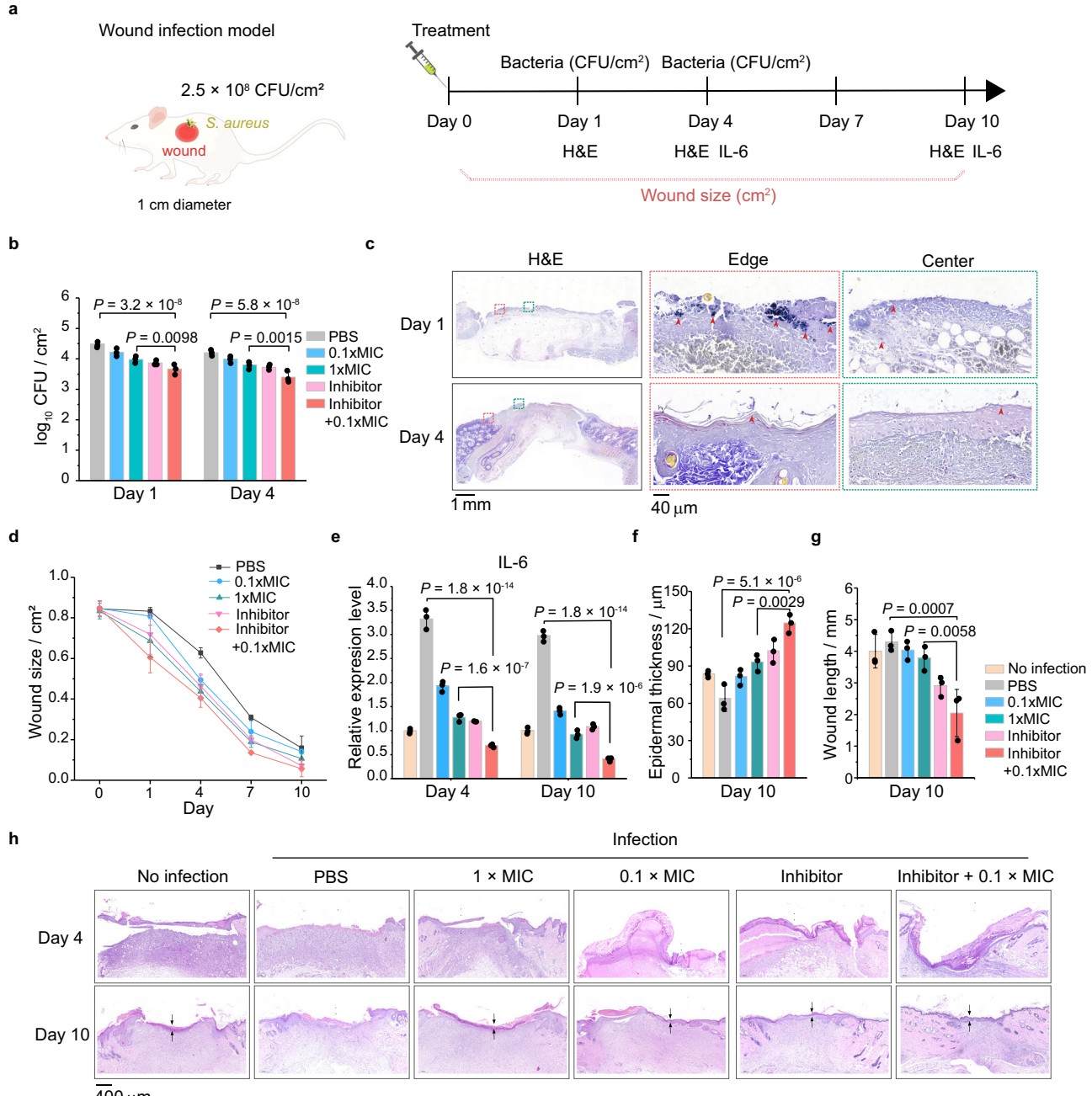

**Fig. 5 | Collagen IV inhibitors as a potential antibiotic adjuvant against bacterial infection in vivo. a** Schematic diagram of the rat wound infection model. **b** Statistical data of *S. aureus* colonies adhering to the wounds quantified by the colony-count technique on days 1 and 4 ($n = 3$). **c** Representative staining images of hematoxylin and eosin (H&E) and Gram in the bacteria infected wounds on days 1 and 4. *S. aureus* were stained black and some of them were indicated by red arrows (Left: scale bars, 1 mm, Middle and Right: scale bars, 40 μm). **d** Statistical data of the wound size over time ($n = 3$). **e** Statistical data of the relative expression level of IL-6

on days 4 and 10 ($n = 3$). **f** Statistical data of the wound epidermal thickness on day 10 ($n = 3$). **g** Statistical data of wound length on day 10 ($n = 3$). **h** Representative staining images of H&E in the wounds on days 4 and 10. Scale bars, 400 μm. All these measured data were shown as mean ± SD ($n = 3$ wounds from 3 independent experiments; one-way ANOVA), unless otherwise stated. All representative data were repeated at least three times with similar results. Source data are provided as a Source Data file.

collagen I (Gibco) for 30 min in the dark at RT. The agarose stamp with the micropatterned side was carefully printed on a clean coverslip to obtain the collagen micropatterns after the excess collagen was removed. The coverslip with collagen micropatterns was placed on the PAAm prepolymer solution, followed by polymerization, thereby transferring the micropatterns onto the PAAm gels. The PAAm gel substrate with micropatterns was sterilized under ultraviolet (UV) irradiation for 30 min after washing three times with PBS. A drop containing approximately 500,000 epithelial cells was seeded on the

PAAm gel substrate for 24 h to form micropatterned cells. In particular, a single IEC-6 cell was seeded on the micropatterned substrate (200 μm diameter) for 4 days to form monoclonal micropatterned cells. Low-density micropatterned cell monolayers were formed by incubating 10%, 20%, 40%, and 60% of the original density (500,000 epithelial cells) of IEC-6 cells on the PAAm gel substrates with specific micropatterns for 24 h, respectively. In some cytochalasin D-related experiments, micropatterned cell monolayers were treated with 10% DMEM containing 1 μg/ml of cytochalasin D for 2 h, washed twice with

PBS, and then infected with *S. aureus* for 2 h or directly fixed with 4 % paraformaldehyde for immunofluorescence staining.

## Cell culture

Rat small intestinal epithelial cell line-6 (IEC-6, ATCC CRL-1592) and human keratinocyte cell line (HaCaT, Cell Bank of the Chinese Academy of Sciences SCSP-5091) were maintained in Dulbecco's Modified Eagle's Medium (DMEM, Gibco) supplemented with 10% fetal bovine serum (FBS, Invitrogen) and 1% Penicillin–Streptomycin solution (Gibco) at 37 °C and 5% $CO_2$. The HaCat cell line purchased from Beijing zhongkezhijian Biotechnology Co. Ltd. had passed the short tandem repeat (STR) test. All bacterial cell infection experiments were performed in antibiotic-free DMEM with 10% FBS.

## Bacterial culture

*Staphylococcus aureus* (ATCC29213) and *Escherichia coli* (ATCC25922) were transfected with pSC19-GFP (erythromycin resistance) to express green fluorescent protein (GFP). *S. aureus* USA300 and Cna-deficient *S. aureus* USA300 were stained with pHrodo (ThermoFisher, P36600). They were grown on Luria-Bertani (LB) liquid medium at 37 °C with shaking at 180 rpm for 10 h.

## Construction of *S. aureus* mutants knocking out the Cna gene

*S. aureus* RN4220, *E. coli* DH5α, and shuttle plasmid pKOR1 (from FORHIGH BIOTECH Co., Ltd.) were used to construct Cna-knockout *S. aureus* USA300 strains based on the reverse screening system for gene editing technology[64]. Briefly, the upstream and downstream DNA fragments of the target gene Cna (Cna-upstream-F 5′-AATCAAG-GACTTCCTGGAGCACT-3′ and Cna-upstream- R 5′-GTATAAGGAGGGG TTTTTACCATTATAATTATTTTTATAG-3′ and Cna-downstream-F 5′- TA AAAAATAATTATAATGGTAAAAACCCCTCCTTATACTCT-3′ and Cna-downstream-R 5′- ATTAGATGTGCTAGATGCTAAAT-3′) were first amplified by polymerase chain reaction (PCR), and then integrated and amplified by fusion PCR. The fusion fragment was cloned into the temperature-sensitive plasmid pKOR1 to obtain a knockout vector, and the positive plasmid was screened from chemical transformation of *E. coli* DH5α recipient cells. Further, the knockout vector was introduced into restriction-deficient *S. aureus* RN4220 for modification, which was subsequently extracted and electrotransformed into *S. aureus* USA300 to replace the Cna gene by allelic integration. They were screened with chloramphenicol and dehydrated tetracycline, and finally identified by PCR. In this way, the Cna-knockout *S. aureus* USA300 was successfully constructed.

## Bacteria/bead-cell interaction model

IEC-6 cells were cultured on the micropatterned PAAm substrates for 24 h. Then, bacteria or yellow–green fluorescently labeled polystyrene beads (SIGMA, L4655) were allowed to interact with the micropatterned IEC-6 cells at 37 °C and 5% $CO_2$ for 2 h. After bacteria–cell infection, the substrate was washed once with PBS to remove non-adherent bacteria.

## Immunostaining

The substrate with infected cells was fixed with 4% paraformaldehyde (PFA) for 20 min and then blocked with 1% bovine serum albumin (BSA, Beyotime) for 1 h at RT. Actin filaments were stained with Actin-Tracker Red-555 (Thermo Fisher, R37112, Two drops per ml) for 30 min, and the nuclei were marked with DAPI for 10 min. Collagen I, collagen II, collagen IV and vinculin on cells were labeled with anti-collagen I (Immunoway, YM3764, 4H10, 1:50), anti-collagen II (Immunoway, YM3749, 1H1, 1:50), anti-collagen IV (Novus Biologicals, NB120-6586, 1:50) and anti-vinculin (Bioss, bs-23650R, 1:50) primary antibodies, respectively, overnight at 4 °C. Alexa Fluor 647-labeled goat anti-mouse secondary antibodies (Beyotime, A0473, 1:200) or Alexa Fluor

488-labeled goat anti-rabbit secondary antibodies (Beyotime, A0423, 1:200) were further incubated in the dark for 1 h.

## Quantification of fluorescence intensity

Quantitative imaging was performed on a confocal laser scanning microscope (CLSM, Nikon). Fluorescence images were analyzed based on specific MATLAB programs to obtain fluorescence intensity heat-maps and radial curves. Briefly, fluorescence images of the same group were stacked together to quantify the mean pixel fluorescence intensity. To compare the fluorescence intensity between the different groups, the mean fluorescence intensity was normalized to obtain heat maps of spatial distribution. The circular micropattern was divided into 100 concentric regions of equal width, and the pixel intensity of each concentric region was calculated to obtain the average fluorescence intensity curves along the radius.

## Characterization of adhesion forces

Fluidic force microscopy (FluidFM, Cytosurge) based on microfluidic technology and atomic force microscopy (AFM) was used to measure the adhesion forces of single cells. According to the size of bacteria and beads (1 μm diameter, Sigma), the probe with circular opening (300 nm diameter) and the spring constant (0.6 N/m) was selected. The cantilever sensitivity (S, m/v) was calibrated by the Sader method[65]. The probe was coated with 0.1 mg/ml PLL-g-PEG for 20 min to diminish nonspecific cell binding, followed by washing in filtered water for 5 min. During the force spectroscopy experiment, a single bacterium was tightly immobilized at the cantilever opening by applying negative pressure (−200 mbar). Then, the cantilever with the single bacterium approached the selected cell at a piezo speed of 1 μm/s and paused 30 s to interact with the cell when the set point (100 nN) was reached. Finally, the cantilever was retracted from the cell at a piezo speed of 1 μm/s to obtain the adhesion force data.

## Single-cell RNA sequencing

ScRNA-seq library construction and sequencing were performed by GeekGene Co., Ltd. Micropatterned IEC-6 cells were covered with 0.25% trypsin (Gibco) for 3 min at RT. Then, the micropatterned cells at the edge and at the center were isolated into tubes with lysis component and ribonuclease by a mouth pipette (30 μm diameter). Total RNA was isolated for amplification using the Smart-seq2 method[39]. The quality of the established library was tested, and the qualified libraries were sequenced. Both the center group and the edge group contained two biological replicates, and each biological replicate contained two cells.

## Treatment of wound infection

A total of 30 60-day-old Sprague-Dawley (SD) female rats were used for wound infection experiments. The SD rats were divided into six groups, including No infection group, PBS group, 0.1 × MIC group, 1 × MIC group, Inhibitor group, and Inhibitor + 0.1 × MIC group. Each group contained three independent samples. Excisional wounds (1 cm in diameter) were created on the left and right backs. The formed wounds were first treated with PBS. Then, they were treated with PBS, *S. aureus* ($2.5 \times 10^8$ CFU/cm², GFP-labeling, erythromycin resistance) and PBS, *S. aureus* and ciprofloxacin (1 μg/mL, 0.1 × MIC), *S. aureus* and ciprofloxacin (10 μg/mL, 1xMIC), *S. aureus* and collagen IV inhibitor (50 μm VU6015929), or *S. aureus* and collagen IV inhibitor (50 μm VU6015929) + ciprofloxacin (1 μg/mL, 0.1 × MIC). Three rats in each group were sacrificed at 1, 4, and 10 days, respectively. The infected tissues were homogenized with saline at a ratio of 1:10, then diluted and spread on the LB ager plates at 37 °C for 24 h to evaluate the number of bacteria in the wounds. The CFUs of adhesive *S. aureus* on days 1 and 4 and the gene expression of interleukin-6 (IL-6) on days 4 and 10 were measured. The wound area of each rat was recorded and

quantified with ImageJ on days 0, 1, 4, 7, and 10. The H&E staining results on days 1, 4, and 10 and Gram staining for *S. aureus* on days 1 and 4 were detected.

## Statistical analysis

All statistical analyses and graphs were performed on Origin Pro 2017. *P values* ($P < 0.05$) were considered statistically significant.

## Reporting summary

Further information on research design is available in the Nature Portfolio Reporting Summary linked to this article.

## Data availability

The scRNA-seq data used in this study are available in the NCBI SRA database under accession code PRJNA997613. The remaining data generated in this study are provided in the Supplementary Information or Source Data file. Source data are provided with this paper.

## Code availability

Codes used in this work are available from the corresponding author on reasonable request.

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

## Acknowledgements

We thank the Core Facilities of the School of Life Sciences and National Center for Protein Sciences at Peking University in Beijing. We thank Prof. Kui Zhu from China Agricultural University for his helpful discussion and suggestions. This work was partially funded by the Natural Science Foundation of Beijing Municipality (Grant No. Z200017 to J.H.), the National Natural Science Foundation of China (Grant Nos. 12372175 and 11972001 to J.H.), and the National Key Research and Development Program of China (Grant No. 2021YFA1000201 to J.H.).

## Author contributions

J.H. conceptualized and designed the work. Y.F. performed the experiments and acquired the data. X.L., Y.H., H.X., W.X., Z.T., Z.Y., Z.W., and L.X. performed the mechanical characterization and animal experiment, and helped to analyze the data. S.W. and X.D. performed the numerical simulations. S.Q. and K.H. helped to analyze the data. Y.F. and J.H. wrote the manuscript, which was proofread and edited by all co-authors prior to submission.

## Competing interests

The authors declare no competing interests.
