## [Peer Review File · Nature Communications]

Geometric constraint-triggered collagen expression mediates bacterial-host adhesionREVIEWER COMMENTS

Reviewer #1 (Remarks to the Author):

In this paper, the authors report that the adhesion of two bacterial species, *E. coli* and *S. aureus*, to host cells depends on increased collagen at the edges of host cell populations, which the authors ascribe to geometric constraints.

Monolayers with controlled shape and size were firstly infected with *S. aureus* and *E. coli* cells and bacteria adhesion evaluated through fluorescence microscopy. Spatially heterogeneous interactions were revealed, with bacterial adhesion being much higher for cells located at the edges of the monolayers than for those located at the center. This behavior was shown to be dependent on the stiffness of the substrates used to culture the cells, i.e., bacterial adhesion on the edges increased with the substrate stiffness. In a second part of the study, the authors performed fluidic force microscopy with bacterial probes to quantify the forces driving the heterogeneous bacterial adhesion. Adhesion forces between single bacteria and cells located at the edges and the center of the monolayers were measured. Higher adhesion forces are reported for cells located at the edges, regardless of monolayer dimensions or the type of bacteria. Adhesion was also shown to depend on contact time and retraction speed. To understand which molecular factors are responsible for the heterogeneous bacterial adhesion observed under geometric constraints, the authors use single-cell RNA sequencing and found that the gene *Col4a1*, encoding collagen IV, is upregulated in cells located on the edges of the monolayers. Monte Carlo simulations (relying on the assumption that the bacteria-cell interactions are mediated by collagen IV) modeled the results of the *in vitro* experiments. Finally, using a rat excised wound infection model, the authors demonstrate that a collagen IV inhibitor works synergistically with other antibiotics *in vivo*.

While this work might be of clinical importance (especially the promising results obtained with the ColIV inhibitor), for it to be truly impactful at a mechanistic level some major issues need to be addressed. At this stage, I am not 100% convinced that it is necessarily the geometric constraints that induce peripheral cell collagen production (see major issues below).

Major issues:

1. Why should it necessarily be geometric constraints that induce the presence of collagen in cells at the edges of the 2D monolayers? Is the presence of collagen on cells at the edges of the 2D monolayers not merely a consequence of them being at the periphery? This would be in line with the observations made by the authors that ultimately the shapes of the 2D monolayers did not influence peripheral collagen expression patterns. To test whether spatial confinement is really a stimulus for collagen expression at the edges, the authors could culture cells under non-geometrically confined conditions and perform their various measurements on microcolonies before confluency. If microcolonies do not show peripheral collagen expression, then there is a case that geometric confinement plays a role.
2. If geometric confinement indeed stimulates collagen production at the edges, the authors should provide some explanation for this. Is it related to the distributions of tensile strain or rather compressive forces within the 2D monolayers? Is it because mechanosensation? The authors should supply data (mechanical, e.g., relying on AFM and transcriptional, e.g., of genes involved in mechanosensation).
3. I am not convinced of the specific involvement of bacterial adhesins in the adhesion observed by the authors:
 - a. The adhesion profiles (Fig. 2c) look like typical profiles for short-range acting electrostatic / hydrophobic adhesion, including the jump to contact and the absence of typical polymeric extension profiles.
 - b. Perhaps the unbinding (retraction) profiles are the sum of many parallel bonds with collagen. Maybe okay for *S. aureus*, whose adhesins are known to bind their ligands with extraordinary mechanical stability. But not for *E. coli* whose adhesins form complexes typically 10 times weaker (not only ~ 2 times weaker).
 - c. To really implicate specific adhesins in binding to the collagen, the authors could perform experiments with strains lacking collagen-binding adhesins.

Minor issues:

1. Although this is not crucial for the study, since the authors present the stiffness values of the three different substrates (e.g. lines 100, 156 and 165), they could mention how these were determined.

2. The authors do not show any data regarding adhesion frequency. It would be interesting to understand if the higher forces registered for the monolayers edges are associated to higher binding frequencies or, on the contrary, the binding frequency is independent on the region being probed. This would provide additional information on the cellular heterogeneity.
3. What are the biological implications of the observed bacterial adhesion dependency on the substrate stiffness?
4. From Fig. 3 b-c, other genes seem to have their expression altered according to their location within the cell monolayers. If possible, could the authors comment on this? Is this expected for this type of cells?
5. In Fig. 3c, the labels "up", "down" and "NS" used for red, blue and gray dots, respectively, are not in agreement with the figure caption. For instance, the gene Col4a1, whose expression is upregulated according to Fig. 3b, appears as a blue dot in Fig. 3c.
6. Fig. 4a. The collagen hug mechanism used by *S. aureus* Cna adhesins is not equivalent to DLL1. There are subtle differences between the two mechanisms.
7. Rat infection results: Despite being quite clear, the results description is perhaps too concise. For instance, the staining images results This could be improved by the authors. For instance, no comments are made on the staining results on day 4 (Fig. 5 c).
8. In the last section, the figure calls must be corrected (lines 354 and 358).

Reviewer #2 (Remarks to the Author):

The authors provide clear evidence and abundant data to demonstrate that at least staph aureus binds to the cells at the outer edge of a monolayer when the cells are grown in a spatially restricted area. They further argue that this finding is not dependent on cell type or bacterial type and show that it does not occur with latex beads. The authors present evidence that, for Staph aureus, the binding is dependent on the increased production of collagen IV from the cells in the outer region of the spatially constricted monolayer. They show using an inhibitor of collagen IV that they can prevent the increased association with bacteria with the outer areas of the spatially constricted monolayer and provide evidence that, in vivo, reduction of collagen IV production enhances the activity of antibiotics. The results are interesting and worthy of the broader readership of Nature Communications, but I have significant concerns about the presentation of the findings, the interpretation of the data, and the controls that are used in the experiments.

Major concerns

1. For several of the presented experimental data sets, inappropriate statistical analyses are used. I.e. Fig. 1F, Fig. 4, Fig. 5 - t-tests are used when there are more than 3 groups, and should instead be a test that better controls for false positive rate.
2. Extended Fig. 1: The authors present data in panel A to argue that, when cells are not spatially restricted, bacteria adhere throughout the monolayer. However, for this reviewer, it is difficult to draw the same conclusion. It appears that the monolayers of cells in A are significantly less dense. Would it not be true that in a spatially restricted setting in which the cells were less confluent the bacteria would be throughout the monolayer as well? I guess I am having trouble with determining whether the finding is spatial restriction per se or just that cells at increased density are bound to less well by bacteria except for cells at the periphery. This is not to say that I don't disagree with the authors findings. We routinely observe that bacterial adherence and invasion is significantly increased in areas of a monolayer that are either less dense or have gaps in confluency.

Could the authors perform the same experiment as presented in panel a with cells that grow as patches. Something like a Caco-2 cell or a T84 cell, or after a couple of days following seeding at a low confluency of IEC-6 cells such that there are patches that form in the absence of a spatially restriction. In this situation, it seems that if *S. aureus* bound better to the outer region of these patches then it seems the interpretation would be that rather than spatial restriction changing the binding pattern of bacteria, it is either the density of the bacteria or the ability to access substrates, such as collagen iV, for binding at the edge of the monolayer.

It is unclear whether in the spatially restricted setting the density is such that any mucus is being produced, this will occur for the cells at a higher density at a greater rate than for cells at the lower density at the edge of the spatial restriction or at a subconfluent state. RNAseq data

presented may have some insights. I am not certain whether the IEC-6 cells even produce mucus.

3. Extended figure 2: could the authors comment on whether the increased amount of bacteria is believed to be new adhesion events or just replication, in some areas in figures presented it appears that *S. aureus* has invaded the cells and may just be replicating, which would produce an increased fluorescent signal without really changing the amount of bacteria that initially bound to the cells.

4. Extended data Fig. 3: could the authors clarify what is meant by "monoclonal IEC-6"? Can the authors tell whether the poly-l-lysine is really enabling the bacteria to adhere to cells or do the bacteria just stick to the poly-l-lysine substrate. We routinely use poly-l-lysine to increase the adhesion of bacteria to cells and to stick bacteria to surfaces in a non-specific manner, no images are presented to say that the bacteria are not just in locations of surface where there are not cells. Do the bacteria invade the cells following the poly-l-lysine mediated adhesion? Given the dependency of the phenotype on Collagen IV and not other collagen types (extended data Fig. 6) it is also a little concerning the poly-l-lysine observation may just be the result of non-specific interactions that are well known.

5. Extended data 5: could the authors indicate on the figure or in the graph what the denominator and numerator are to calculate the foldchange.

6. Extended Fig. 7: Here the authors try to conclude that the bacteria lose the ability to differentially bind to different regions of the monolayer. Aren't they really just demonstrating that collagen IV is necessary for binding. They show cells in the middle of the monolayer lack collagen IV expression, and the bulk of collagen IV present is likely below the cells in the interior of the monolayer where it would be unavailable to the bacteria for binding?

7. Extended Fig. 9: The fur is blocking some of the wounds, how was this controlled for? For the measurements. I.e. Were the measurements done on the animal or from the images presented such that the presence of the fur may make the measurements less reliable.

8. Fig. 4: Monte Carlo simulation, unclear from table S2 how the beads were handled to show they result in less spatially confined binding in the spatially restricted monolayer.

9. Fig. 5: The data presented in the histology are not clear. Particularly panel C, which is at such a low magnification that no bacteria can be observed to support the authors conclusion about the location and or abundance of bacteria in the tissue section. The n=3 is vague. Is this 3 animals or three experiments? If three experiments how many animals were used? The animal number is unclear in the methods and figure legends. The authors ignore the impact of reducing collagen on the ability of immune cells to access the tissue site and control the infection. They also ignore the possibility of greater tissue penetration by the antibiotic as alternative mechanisms rather than reduced adhesion. More concerning is what is the impact of the tyrosine kinase inhibitor on the immune responses measured in the study. This isn't to say that the inhibitor+antibiotic are more not effective than the antibiotic alone, but that it remains unclear to this reviewer whether the impact of the inhibitor is on fewer bacteria binding to the cells or a number of other factors in vivo.

Minor concerns:

1. Dots are used in many of the graphs throughout the manuscript. They are too small to be seen.

2. Line 26-27: Whereas some bacteria attach and invade cells, all bacteria do not invade cells following adherence. The sentence should be clarified.

3. Line 40: Whereas intracellular bacteria can modify mechanotransduction, extracellular bacteria can as well. I would clarify the statement.

4. Line 43: Sca4 is produced by *Rickettsia parkeri* not *Parkerella*

5. Figure 2: could you make the Y-axis labels consistent for easier comparison in panels h and i.

Reviewer #3 (Remarks to the Author):

Manuscript Summary:

In this manuscript the authors show that cells in geometrically confined environments exhibit strong spatial heterogeneity, including heterogeneous gene expression, which has a critical impact on adhesion of bacterial pathogens onto host cells. Specifically, they find that increased expression of collagen IV for cells located at edges as opposed to centers of micropatterned monolayers, leads

to increased adhesion of bacteria due to enhanced adhesion forces onto the host cell surface. Moreover, this phenomenon is matrix stiffness-dependent and accentuated for cells residing on stiffer matrices. Through a Monte Carlo simulation the authors were able to recapitulate the in vitro findings in silico and, though the usage of an in vivo model they were able to show that collagen IV inhibitors increase antibiotic efficacy. Overall, I think this work is novel and interesting in that it provides new insights into how host cells interact with infectious bacteria depending on their geometrical location and how mechanical adhesion forces play a key role on those interactions. However, I have some concerns about data presentation and analysis, overstatements, and some omissions/unclear points throughout the manuscript.

General Comments:

(1) In the Introduction there are several erroneous statements that need to be corrected. For example, authors state: "For example, the virulence factor Sca4 secreted by *Parkerella* specifically binds to some cell-adhesion proteins (e.g., vinculin) to reduce intercellular tension, allowing bacteria to spread more easily by manipulating cytoskeletal forces." First mistake, there is no bacterium called *Parkerella*. The intracellular bacterial pathogen in this study was *Rickettsia parkeri*. Second mistake: Sca4 does not bind to some cell-adhesion proteins. In this study it was shown that Sca4 specifically binds to vinculin, not allowing vinculin to bind to its binding partner α -catenin thus disrupting the organization of adherens junctions. Also later on the authors state: "In contrast, collective responses of epithelial monolayers to bacterial invasion can trigger the extrusion of infected cells to limit local spread of infection". This study did not look at bacterial invasion whatsoever, but rather at intercellular spread of bacteria through host cells during the course of hours/days post-infection. Thus, this statement also needs to be corrected.

(2) In the last paragraph of the introduction the authors state: "With the aid of well-established microcontact printing technique, we first develop an in vitro model of bacteria infecting cells, which allows pathogenic bacteria to infect host cell monolayers grown on spatially confined extracellular substrates. We demonstrate that there are remarkably spatially heterogeneous bacteria-host interactions owing to the presence of two-dimensional (2D) geometric constraints on host monolayers." The authors only tested two different bacterial pathogens, namely *S. aureus* and *E. coli*, infecting just two cell lines, namely HACAT and IEC6. Statements like the above, sound like overstatements and misleading. It looks like the authors findings apply to all cell lines and bacterial pathogens, which is not the case or at least has not been tested. So, if they want to keep the general statements, they would have to test a lot more pathogens and see if same findings apply. Otherwise, I would introduce from the introduction the model pathogens tested and give some background as to why those were chosen to be tested. Along that line it looks like "Collagen IV" is an important receptor for adhesion of *E. coli* and *S. aureus* and that would be nice to be introduced somehow, since it is something already known. Also, is it an important adhesion receptor for more bacterial pathogens? Finally, the statement in the discussion "indicating that the absolute magnitude of bacterial adhesion forces were largely dependent on the specific type of bacteria" it is clearly an overstatement since only two bacterial pathogens were tested. This has to either be more specific or experiments with more bacterial pathogens have to be conducted.

(3) A general very important comment: Could it just be the cell shape and not the geometric constraint per se that drives the differential adhesion of bacteria onto host cells? I am wondering how much the geometric constraints are responsible for the spatially heterogeneous bacteria-cell adhesion pattern as compared to the cell shape. Can the authors use the F-actin signal (since it looks in Figure 1 that it localizes at the cell-cell edges) or some cell-cell edge marker (e.g., anti-E-cadherin antibody) to segment cells and figure out how the shape of cells at the edges differs from the shape of cells at the center? Could it be that the distinct shape of the cells is what governs changes in gene expression and thus bacterial pathogen adhesion onto host cells? Could it be that on soft matrices cells at the edges are more similar to cells at the center (as opposed to stiff) and that is why the heterogeneity of bacterial adhesion is decreased? In line with that if one looks at the images in Figure 1D, it looks like on the soft hydrogel the cells are more homogeneously spread, or that cells at edges look more like cells in the middle. This is not the case on stiff hydrogel. So, is there a way of assessing the role of cell shape on modulating bacterial adhesion (potentially independent of geometric constraint)?

(4) In the legend of Figure 1 it is stated: "All these measured data were shown as mean \pm SD

($n=5$; two-tailed unpaired t-test), unless otherwise stated.” What does $n=5$ mean? Five independent hydrogels? Or five independent islands within the same hydrogel? I think if it is five independent islands (i.e., just five images) definitely more images need to be taken on hydrogels prepared on independent days, since the variability in stiffness of polyacrylamide hydrogels is well documented. The same question applies to figures shown later on (e.g. Figure 3). Also, for comparing distributions authors should probably better use a non-parametric test rather than a t-test (like Wilcoxon ranksum). Also, why did the authors chose 10 kPa, 32 kPa and 93 kPa? Is there any (patho)physiological relevant behind those chose values of matrix stiffness? This should be explained.

(5) Authors state: “To further assess the generality of our findings, we changed the geometric constraint parameters of micropatterned cell monolayers, including their size (100, 200 and 400 μm diameter). Under these conditions, the spatially heterogeneous adhesion phenomenon remained unchanged, indicating that the bacteria–host adhesion behavior was independent of the size of geometric constraints”. This is a bit confusing because if one looks at Figure 1c it is clear that the larger the diameter, the bigger the difference in adhesion between center and edge cells. If one compares $D=100\ \mu\text{m}$ versus $D=400\ \mu\text{m}$, the relative difference in the latter case ($D=400$) is at least four times bigger than in the former ($D=100$). So I think it is not fair to claim that the diameter does not play a role (when it does) but then say that the matrix stiffness plays a role when the relative increase on stiff matrix as compared to soft is approximately three-fold if one looks at the boxplots of Figure 1f. Back again to the comment #3 I made, I speculate that the shape of cells at edges versus center might play a key role on the observations made above.

(6) Do higher adhesion forces and higher adhesion suggest higher host cell invasion? Previously it was shown *Listeria monocytogenes* adheres more on endothelial cells residing on stiffer extracellular matrices as opposed to softer, and that ultimately to more bacterial invasion (1)? I think this experiment might be very interesting to perform and relatively easy with some differential immunostaining of the previously fixed samples or with confocal imaging of the current ones.

Minor comments:

(1) In figure 4 authors show how matrix stiffness impacts the expression of collagen IV. What about if one was to compare $D=100$ to $D=400\ \mu\text{m}$ islands? Is the difference going to be smaller for $D=100$ as expected?

(2) Because of color blind people use of red and green (especially overlay) should be avoided. For example, see immunofluorescence images in Figure 1. Maybe change the red color to magenta?

(3) In the boxplots used in Figure 2 h, Figure 3 g etc it would be nice to add the individual data points pretty much as authors do for Figure 1 at the barplots. That would increase transparency and give a sense of the variability between experiments.

(4) In Figure 2G why are beads only and *S. aureus* shown and not *E. coli*?

Reference

1. E. E. Bastounis, Y.-T. Yeh, J. A. Theriot, Matrix stiffness modulates infection of endothelial cells by *Listeria monocytogenes* via expression of cell surface vimentin. *Molecular Biology of the Cell* 29, 1571-1589 (2018).

Point-by-point responses to the reviewers' comments and suggestions

(Manuscript Number: NCOMMS-23-11807A)

Reviewer #1:

In this paper, the authors report that the adhesion of two bacterial species, E. coli and S. aureus, to host cells depends on increased collagen at the edges of host cell populations, which the authors ascribe to geometric constraints. Monolayers with controlled shape and size were firstly infected with S. aureus and E. coli cells and bacteria adhesion evaluated through fluorescence microscopy. Spatially heterogeneous interactions were revealed, with bacterial adhesion being much higher for cells located at the edges of the monolayers than for those located at the center. This behavior was shown to be dependent on the stiffness of the substrates used to culture the cells, i.e., bacterial adhesion on the edges increased with the substrate stiffness. In a second part of the study, the authors performed used fluidic force microscopy with bacterial probes to quantify the forces driving the heterogenous bacterial adhesion. Adhesion forces between single bacteria and cells located at the edges and the center of the monolayers were measured. Higher adhesion forces are reported for cells located at the edges, regardless of monolayer dimensions or the type of bacteria. Adhesion was also shown to depend on contact time and retraction speed. To understand which molecular factors are responsible for the heterogeneous bacterial adhesion observed under geometric constraints, the authors use single-cell RNA sequencing and found that the gene Col4a1, encoding collagen IV, is upregulated in cells located on the edges of the monolayers. Monte Carlo simulations (relying on the assumption that the bacteria-cell interactions are mediated by collagen IV) modeled the results of the in vitro experiments. Finally, using a rat excised wound infection model, the authors demonstrate that a collagen IV inhibitor works synergistically with other antibiotics in vivo.

While this work might be of clinical importance (especially the promising results obtained with the ColIV inhibitor), for it to be truly impactful at a mechanistic level some major issues need to be addressed. At this stage, I am not 100% convinced that it is necessarily the geometric constraints that induce peripheral cell collagen production (see major issues below).

Response: We sincerely thank the reviewer's positive comments and constructive suggestions, which are very helpful to revising our manuscript. In order to make it clearer, more complete,

and more readable, we have accordingly supplemented some necessary experiments, statements and references in the revised manuscript, and simultaneously clarified the reviewer's concerns.

Major comments:

1. Why should it necessarily be geometric constraints that induce the presence of collagen in cells at the edges of the 2D monolayers? Is the presence of collagen on cells at the edges of the 2D monolayers not merely a consequence of them being at the periphery? This would be in line with the observations made by the authors that ultimately the shapes of the 2D monolayers did not influence peripheral collagen expression patterns. To test whether spatial confinement is really a stimulus for collagen expression at the edges, the authors could culture cells under non-geometrically confined conditions and perform their various measurements on microcolonies before confluency. If microcolonies do not show peripheral collagen expression, then there is a case that geometric confinement plays a role.

Response: Thanks for your instructive comments and suggestions. Based on your suggestions, we cultured low-density IEC-6 cells (10%, 20%, 40% and 60% of 500,000 cells, respectively) on stiff hydrogels with circular micropatterns or 12-well plates for one day, and subsequently performed immunofluorescence staining for collagen IV in the cell monolayers under non-geometrically constrained conditions where these cells were not fused. We found that there was no significant peripheral collagen IV expression in the circular micropatterned cell monolayers under non-geometric constraints (Supplementary Fig. 6d). However, when the IEC-6 cells were confluent to the geometric limit, the collagen IV expression increased at the edges of the micropatterned cell monolayers, thus creating the edge effects (Supplementary Fig. 6d and Fig. 3d). In addition, for the cell monolayers cultured in 12-well plates without geometric constraints, one could observe that peripheral expression of collagen IV was consistently absent (Supplementary Fig. 6e). These results implied that spatially geometric constraints induced collagen production in the cells at the edges of a 2D constrained monolayers. (See Page 8, Line 6-12 in the manuscript; Page 14, Line 17-19 (Methods) in the manuscript; Supplementary Fig. 6d-e in the Supplementary Information)

2. If geometric confinement indeed stimulates collagen production at the edges, the authors should provide some explanation for this. Is it related to the distributions of tensile strain or rather compressive forces within the 2D monolayers? Is it because mechanosensation?

The authors should supply data (mechanical, e.g., relying on AFM and transcriptional, e.g., of genes involved in mechanosensation).

Response: Thank you very much for your professional comments. Based on your useful suggestions, we further explored the reasons for the geometric constraints to stimulate collagen production at the edges of 2D cell monolayers. From the Kyoto Encyclopedia of Genes and Genomes (KEGG) results of a single-cell RNA sequencing, we noticed that focal adhesion pathway highly associated with cellular traction forces was significantly upregulated at the edge of micropatterned cell monolayers (Sarangi et al. 2017), where cellular traction forces were shown to be closely dependent on ECM stiffness and bacterial infection (Bastounis et al. 2022; Liu et al. 2021) (Supplementary Fig. 5b). Therefore, we first performed immunofluorescence staining for vinculin, which was associated with the focal adhesion pathway and involved in mechanochemical signalling (Elosegui-Artola et al. 2016), and verified that the vinculin expression was also increased at the edges of the micropatterned cell monolayers (Supplementary Fig. 8a). Moreover, we further quantified the spatial distribution of cellular traction forces in the 2D micropatterned cell monolayers cultured on the substrates (Supplementary Fig. 8b-c). We found that the cellular traction forces at the edges of the micropatterned cell monolayers was significantly greater than those at the central regions, and that the cellular traction forces in the cell monolayers on the relatively stiff substrates were greater than those in the cell monolayers on the soft substrates, which were highly consistent with the distributions of collagen IV expression in the cell monolayers under geometric constraints, suggesting that cellular traction forces might modulate the distributions of collagen IV expression in the geometrically constrained cell monolayers.

To further validate the role of cellular traction forces in the collagen IV expression at the edge of the micropatterned cell monolayers, we employed cytochalasin D to reduce the cytoskeletal forces in the IEC-6 cell monolayers under geometric constraints. After treating circular micropatterned cell monolayers with cytochalasin D for 2 h, one could readily observe that collagen IV expression and adhesive bacteria at central and peripheral locations no longer had spatial heterogeneity, indicating that cell mechanosensation might play an important role in regulating collagen IV production and its spatial distribution in the geometrically constrained cell monolayers (Supplementary Fig. 8d-e). Moreover, based on the reviewer's suggestion, we measured the Young's moduli of cells near the center and edge of the micropatterned cell monolayer with atomic force microscope (AFM). Interestingly, we demonstrated that the

Young's moduli of the cells at the edges were much lower than those at the central regions. This situation was highly correlated with the distribution of collagen IV expression (Chen et al. 2014) (Supplementary Fig. 8f-h). The manuscript have been revised accordingly (See Page 8, Line 34-44 and Page 9, Line 1-6 in the manuscript; Page 14, Line 19-22 (Methods) in the manuscript; Page 14, Line 13-40 and Page 15, Line 1 (Methods: Quantification of cellular traction forces in micropatterned cell monolayers; Quantification of Young's moduli of micropatterned cell monolayers with atomic force microscopy) in the Supplementary Information; Supplementary Fig. 5b and Supplementary Fig. 8 in the Supplementary Information)

3. I am not convinced of the specific involvement of bacterial adhesins in the adhesion observed by the authors:

a. The adhesion profiles (Fig. 2c) look like typical profiles for short-range acting electrostatic / hydrophobic adhesion, including the jump to contact and the absence of typical polymeric extension profiles.

Response: Thank you for your comments and suggestions. In the manuscript, we actually only showed the fitted curves because the fitted curves could make us estimate the values of F_{ad} more easily. As pointed out by the reviewer, the actual curves indeed had some fluctuations in the bacteria-cell adhesion process (As displayed in the below figure). In the revised version, we supplemented the original force curves (See Fig. 2c in the revised manuscript or Fig. R1). In addition, the relevant original data have also been summarised in the source data.

Fig. R1. Typical time-related adhesion force curves between a single bacterium and host cells located near the center and edge of a micropatterned monolayer, which were obtained through the FluidFM-based SCFS. The adhesion force was calculated by quantifying the difference between the lowest point in a specific adhesion force curve and the corresponding one where the adhesion force finally reached a steady state with time.

b. Perhaps the unbinding (retraction) profiles are the sum of many parallel bonds with collagen. Maybe okay for S. aureus, whose adhesins are known to bind their ligands with extraordinary mechanical stability. But not for E. coli whose adhesins form complexes typically 10 times weaker (not only ~2 times weaker).

Response: We sincerely appreciate the valuable comments. In essence, the measurement of adhesion forces between individual bacteria and cells based on FluidFM involves the complex single-cell force spectroscopy (SCFS) rather than the single molecular force spectroscopy (SMFS), even though SCFS and SMFS share many similar characteristics. In general, one doesn't consider the influence of the related substrates in the SMFS analysis. In contrast, in the SCFS analysis, it is likely that the quantification of cellular adhesion forces may also be affected by cellular elasticity, cortical tension, non-specific forces (electrostatic forces and hydrogen bonds) and other factors (Helenius et al. 2008), besides the specific binding between bacterial collagen adhesin and cellular collagen. Additionally, the FluidFM probe usually adsorbs to middle position of individual *E. coli*, which allows for a relatively large contact area between the *E. coli* and the underlying IEC-6 cells. In this situation, it is likely that, in comparison with *S. aureus*, *E. coli* can form more ligand-receptor bonds with the mammalian cells, which may also account for the high adhesion forces.

c. To really implicate specific adhesins in binding to the collagen, the authors could perform experiments with strains lacking collagen-binding adhesins.

Response: Thank you very much for the professional comments. Based on the reviewer's suggestion, we have knocked out bacterial collagen adhesion (Cna) in *S. aureus* using the gene knockout technology to further validate the role of specific collagen adhesin during bacteria-cell heterogeneous adhesion. Since it was quite difficult to be knocked out bacterial Cna in the *S. aureus* ATCC29213 because of chloramphenicol resistance, we thus selected *S. aureus* USA300 as the wild bacterium (wild) and successfully constructed *S. aureus* USA300 with knockout Cna gene (Δ Cna), as re-validated by polymerase chain reaction (PCR). Next, we infected the micropatterned cell monolayers cultured on the stiff substrates using *S. aureus* USA300 (wild) and *S. aureus* USA300 with knockout Cna gene (Δ Cna), respectively. Our experimental results showed that the wild *S. aureus* adhered to the peripheral locations of the cell monolayers with high collagen IV expression, while the Cna-deficient *S. aureus* adhered

less and were relatively homogeneous in spatial locations (Supplementary Fig. 7g-h). Moreover, the spatially heterogeneous adhesion phenomenon presented by the wild-type *S. aureus* USA300 was consistent with that shown by the *S. aureus* ATCC29213 that was used in our previous experiments (Supplementary Fig. 7g). At the same time, we further quantified the adhesion forces of *S. aureus* USA300 (wild type) and *S. aureus* USA300 with knockout Cna gene (Δ Cna) to the micropatterned cell monolayers at the central and peripheral locations, respectively. It could be found that the wild-type *S. aureus* USA300 exhibited much higher bacterial adhesion forces at the peripheral position than the central position of the micropatterned cell monolayers. However, the adhesion forces of *S. aureus* USA300 with knockout Cna gene (Δ Cna) between the central and peripheral positions of the micropatterned cell monolayers were relatively low and not significantly different in our experiments (Supplementary Fig. 7i). These results demonstrated the essential contribution of the specific adhesins bound to collagen in the bacteria-cell interactions and thus further validated that marginal expression of collagen IV triggered by spatially geometric constraints was responsible for heterogeneous bacterial adhesion, as reported in our manuscript. These experimental results have been added to the revised version (See Page 8, Line 24-34 in the manuscript; Page 14, Line 31-42 and Page 15, Line 1-8 (Methods: Construction of *S. aureus* mutants knocking out the Cna gene) in the manuscript; Supplementary Fig. 7g-i in the Supplementary Information)

Minor issues:

1. Although this is not crucial for the study, since the authors present the stiffness values of the three different substrates (e.g. lines 100, 156 and 165), they could mention how these were determined.

Response: In the latest version of the manuscript, we have described how to determine the Young's moduli of these three substrates with different rigidities. In our experiments, the Young's moduli of the soft, medium and stiff substrates were 10.14, 32.29 and 93.46 kPa, respectively, which were all within the stiffness range of biological tissues *in vivo* (Janmey et al. 2020). All these polyacrylamide (PAAm) substrates were prepared based on some previous studies (Denisin and Pruitt 2016) and their Young's moduli were quantified via a Piuma nanoindenter equipped with a Chiaro indenter head (Optics11 Life, Netherlands), as described in the section of “**Characterization of PAAm hydrogel stiffness**” in the Supplementary Information of the revised manuscript (Supplementary Fig. 1e-f). (See Page 4, Line 6-8 in the

2. The authors do not show any data regarding adhesion frequency. It would be interesting to understand if the higher forces registered for the monolayers edges are associated to higher binding frequencies or, on the contrary, the binding frequency is independent on the region being probed. This would provide additional information on the cellular heterogeneity.

Response: Thank you for your suggestions. We agree that exploring binding frequency would be really interesting to further investigate the spatiotemporal dynamics of bacteria-cell adhesion. As described in the above-mentioned section, however, the FluidFM-based single cell force spectroscopy (SCFS) is different from the well-developed single-molecule force spectroscopy (SMFS) to a certain extent. In essence, SCFS is dedicated to the study on the physical/mechanical interactions at the single cell/bacterial level. In this sense, the measurement accuracy and resolution of SCFS is much lower than those of SMFS. Besides, it doesn't consider the effect of the related substrates in the SMFS analysis. In comparison, it is likely that the quantification of cellular adhesion forces may also be affected by cellular elasticity, cortical tension, non-specific forces (electrostatic forces and hydrogen bonds) and other factors (Helenius et al. 2008). As a matter of fact, we could observe some significant fluctuations during the FluidFM-based SCFS measurement, as shown in the figure below. But, it probably does not strictly reflect changes in the adhesion frequency.

Fig. R1. Typical time-related adhesion force curves between a single bacterium and host cells located near the center and edge of a micropatterned monolayer, which were obtained through the FluidFM-based SCFS. The adhesion force was calculated by quantifying the difference between the lowest point in a specific adhesion force curve and the corresponding one where the adhesion force finally reached a steady state with time.

3. What are the biological implications of the observed bacterial adhesion dependency on the substrate stiffness?

Response: Thank you for your thoughtful question. Because there are distinct rigidities for various tissue in human body, such skin and intestines (Janmey et al. 2020). Also, it was found that some pathological conditions, such as atherosclerosis and cancer (Bastounis et al. 2018), could induce changes in extracellular matrix stiffness *in vivo*. Hence, it would be crucial for investigating the potential role of extracellular matrix stiffness in regulating the bacterial infection in human body. As revealed in our current experiments, mechanical aspects of extracellular matrices, such as extracellular matrix stiffness, indeed played a key role in affecting bacterial adhesion as well. This implied that extracellular matrix stiffness might play as important a role as biochemical factors in mediating bacteria-cell adhesion and infection. The possible influence of mechanical cues might also need to be considered in future antimicrobial drug screening and bacterial infection treatment.

4. From Fig. 3 b-c, other genes seem to have their expression altered according to their location within the cell monolayers. If possible, could the authors comment on this? Is this expected for this type of cells?

Response: Thank you for the instructive suggestions. In the single-cell RNA sequencing results, we also found significant upregulation of some genes regulating cell growth and proliferation (Sfrp2, Fam83d) and some genes associated with inflammation (Tnfaip6, Ccl20) at the edge of micropatterned cell monolayers (de Castro et al. 2021; Wang et al. 2015; Dyer et al. 2014; Schutyser et al. 2000). In addition, some genes involved in the focal adhesion pathways, such as Itga5, Itga11 and Parvb, which regulate intracellular actin cytoskeleton dynamics during these processes like cell adhesion, migration, proliferation and survival, were also significantly expressed in the samples located at the edge of cell monolayers (Sarangi et al. 2017; Hamidouche et al. 2009; Sepulveda and Wu 2006). Generally, the focal adhesion pathway is closely related to cellular traction forces, which in part accounts for the mechanical heterogeneity of cell monolayers at the central and peripheral locations caused by the geometric constraints (Sarangi et al. 2017). Similarly, we found that the gene (Tspan2) that mediates signal transduction associated with cell development, growth and motility, was significantly expressed at the central location of the micropatterned cell monolayers (Otsubo et al. 2014). In addition,

some genes related to antimicrobial defence and immune functions (Duox2, Mylk2 and Usp20) were highly expressed, which may explain why the centrally located cells were less susceptible to bacterial infection to some extent (Hong et al. 2016; Wang et al. 2022; Zhang et al. 2019).

To further verify whether there exist similar gene expression phenomena in other cells, we subsequently performed the immunofluorescent staining for collagen IV and vinculin proteins on micropatterned A549 cell monolayers (human non-small cell lung cancer cells, ATCC CRM-CCL-185), which revealed a similar edge effect on the cell monolayers, as shown in Fig. R2. These experimental data suggest that geometric constraint-mediated heterogeneous expression of collagen IV may also occur in other cell monolayers with edge constraints. (See our single-cell RNA sequencing data stored in the NCBI SRA database with accession numbers of PRJNA997613 for more details)

Fig. R2. a, Immunofluorescence staining results and the corresponding distribution heatmaps of collagen IV expression in the A549 cell monolayers with circular micropatterns. **b**, Immunofluorescence staining results and the corresponding distribution heatmaps of vinculin expression in the A549 cell monolayers with circular micropatterns.

5. In Fig. 3c, the labels “up”, “down” and “NS” used for red, blue and gray dots, respectively, are not in agreement with the figure caption. For instance, the gene *Col4a1*, whose expression is upregulated according to Fig. 3b, appears as a blue dot in Fig. 3c.

Response: We sincerely thank you for pointing out these. Based on your suggestions, we have described the figure and the corresponding figure caption in Fig. 3c in more details. The red, blue and gray dots in Fig. 3c represent the genes whose expression levels are up-regulated,

down-regulated and no significant in the central samples compared to the edge samples, respectively. In other words, the red dots represent the genes that are relatively highly expressed in the central regions, the blue dots denote the genes that are relatively highly expressed in the edge regions, and the gray dots indicate the genes whose expression levels have no significant difference between the central and edge regions. Besides, the relative gene expression levels are completely displayed in Fig. 3b, where the red dots stand for the highly expressed genes whereas the blue ones denote the low expressed genes. The corresponding changes have been made in the revise manuscript. (See Fig. 3c in the latest manuscript)

6. Fig. 4a. The collagen hug mechanism used by S. aureus Cna adhesins is not equivalent to DLL! There are subtle differences between the two mechanisms.

Response: We sincerely appreciate the valuable comments and suggestions. We have already corrected the contents in the revised manuscript regarding the collagen hug mechanism and DLL mechanism of *S. aureus* Cna adhesins. (See Page 9, Line 19-25 in the manuscript)

7. Rat infection results: Despite being quite clear, the results description is perhaps too concise. For instance, the staining images results. This could be improved by the authors. For instance, no comments are made on the staining results on day 4 (Fig. 5 c).

Response: Thank you so much for the comments and suggestions. Following your suggestions, we have added a detailed description of the rat infection results. In addition, we have added details of the corresponding methods and numbering for animal experiments in the Methods section. (See Page 11, Line 30-39 in the manuscript; Page 16, Line 11-22 (Methods: Treatment of wound infection) in the revised manuscript)

8. In the last section, the figure calls must be corrected (lines 354 and 358).

Response: Thank you for pointing out the mistakes. We have accordingly corrected the figure calls in the manuscript. (See Page 11, Line 39-43 in the revised manuscript)

Reviewer #2:

The authors provide clear evidence and abundant data to demonstrate that at least staph aureus binds to the cells at the outer edge of a monolayer when the cells are grown in a spatially restricted area. They further argue that this finding is not dependent on cell type or bacterial type and show that it does not occur with latex beads. The authors present evidence that, for Staph aureus, the binding is dependent on the increased production of collagen IV from the cells in the outer region of the spatially constricted monolayer. They show using an inhibitor of collagen IV that they can prevent the increased association with bacteria with the outer areas of the spatially constricted monolayer and provide evidence that, in vivo, reduction of collagen IV production enhances the activity of antibiotics. The results are interesting and worthy of the broader readership of Nature Communications, but I have significant concerns about the presentation of the findings, the interpretation of the data, and the controls that are used in the experiments.

Response: We sincerely thank the reviewer's positive comments and constructive suggestions, which are very helpful to revising our manuscript. To make it clearer, more complete, and more readable, we have accordingly supplemented some necessary experiments, statements and references in the revised manuscript, and simultaneously clarified the reviewer's concerns.

Major concerns:

1. For several of the presented experimental data sets, inappropriate statistical analyses are used. i.e., Fig. 1F, Fig. 4, Fig. 5 - t-tests are used when there are more than 3 groups, and should instead be a test that better controls for false positive rate.

Response: Thank you very much for the professional comments. In the revised manuscript, we have already corrected all the inappropriate statistical analyses. We have conducted appropriate one-way ANOVAs for all these data involving more than three groups, e.g., Fig. 1f, Fig. 4e, h and Fig. 5b, e-g. In addition, all source data have been provided in the resubmitted files. (See Fig. 1f, Fig. 4e, h and Fig. 5b, e-g in the latest manuscript)

2. Extended Fig. 1: The authors present data in panel A to argue that, when cells are not spatially restricted, bacteria adhere throughout the monolayer. However, for this reviewer, it is difficult to draw the same conclusion. It appears that the monolayers of cells in A are

significantly less dense. Would it not be true that in a spatially restricted setting in which the cells were less confluent the bacteria would be throughout the monolayer as well? I guess I am having trouble with determining whether the finding is spatial restriction per se or just that cells at increased density are bound to less well by bacteria except for cells at the periphery. This is not to say that I don't disagree with the authors findings. We routinely observe that bacterial adherence and invasion is significantly increased in areas of a monolayer that are either less dense or have gaps in confluency.

*Could the authors perform the same experiment as presented in panel a with cells that grow as patches. Something like a Caco-2 cell or a T84 cell, or after a couple of days following seeding at a low confluency of IEC-6 cells such that there are patches that form in the absence of a spatial restriction. In this situation, it seems that if *S. aureus* bound better to the outer region of these patches then it seems the interpretation would be that rather than spatial restriction changing the binding pattern of bacteria, it is either the density of the bacteria or the ability to access substrates, such as collagen IV, for binding at the edge of the monolayer.*

It is unclear whether in the spatially restricted setting the density is such that any mucus is being produced, this will occur for the cells at a higher density at a greater rate than for cells at the lower density at the edge of the spatial restriction or at a subconfluent state. RNAseq data presented may have some insights. I am not certain whether the IEC-6 cells even produce mucus.

Response: Thank you very much for your professional comments and suggestions, which is very helpful for us to better revise the manuscript. Following your comments and suggestions, we first carried out experiments as done in the original Supplementary Fig. 1a, where we increased the density of IEC-6 cells under non-geometric constraints. In this situation, we found that there was no edge effect for bacteria-host cells, as shown in the following Fig. R3, suggesting that the spatial distribution of bacterial adhesion was less dependent on cell density in the absence of geometric constraints. Of course, when the cells were not fully confluent, the bacteria could adhere uniformly throughout the underlying monolayers, as the cells were not geometrically constrained at this stage. Further, we found that, in the absence of geometric constraints, the denser the cells grew, the less the cells were infected by bacteria, as pointed out by the reviewer. However, this situation is likely to be different in the presence of geometric constraints on host cell monolayers. In fact, one could frequently observe that the cells near the edge of the micropatterned cell monolayers were prone to bacterial adhesion and infection, but

their local densities might be slightly higher than those in the central regions of the micropatterned cell monolayers, as shown in Fig. 1a and Fig. 1d in the revised manuscript.

Afterwards, we cultured low-density IEC-6 cells for three days, allowed them to form patches without spatial constraints, and then infected them with *S. aureus* for 2 h, as done in our previous experiments (Supplementary Fig. 1b-c). In this situation, we noticed that *S. aureus* adhered relatively uniformly to the surfaces of the patches, regardless of the density and size of the patches, which further demonstrated that spatial geometric constraints are critical for spatially heterogeneous bacteria-host cell adhesion. Finally, we comprehensively screened all differentially expressed genes in single-cell RNA sequencing results once again, which displayed that there were no differential genes associated with mucus in the IEC-6 cells (See also our RNA sequencing data stored in the NCBI SRA database with accession numbers of PRJNA997613 for more details). The relevant experimental data have been added in the revised version (See Page 2, Line 35-39 in the manuscript; Supplementary Fig. 1b-c in the Supplementary Information)

Fig. R3. Representative images of high-density IEC-6 cell monolayers without geometric constraints infected by *S. aureus* expressing GFP.

3. Extended figure 2: could the authors comment on whether the increased amount of bacteria is believed to be new adhesion events or just replication, in some areas in figures presented it appears that S. aureus has invaded the cells and may just be replicating, which would produce an increased fluorescent signal without really changing the amount of bacteria that initially bound to the cells.

Response: Thanks for the useful advice. To clarify whether the increase in the number of adherent bacteria is the result of a new adhesion event or replication event, we characterized the proliferation of *S. aureus* in the context of bacteria-cell interactions. Since the co-culture time for the bacteria-cell adhesion and infection experiments was ~2 hours, we here quantified the colony forming units (CFUs) of *S. aureus* in DMEM over two hours. It was found that

bacterial populations multiplied very little at 40 minutes and increased by 70% over time at 120 minutes (Fig. R4). Therefore, the number of bacteria proliferating should be relatively low during the initial phase of the bacteria-cell interactions. Over time, some bacteria began to proliferate. However, the probability of bacterial replication should be equal whether the bacteria adhered to the center or the edge of the cell monolayers. Overall, the fact that bacteria tend to adhere to and invade the edges of the micropatterned cell monolayers under geometric constraints is well established.

Fig. R4. a-b, Representative images (a) and statistical results of CFUs (b) for *S. aureus* cultured in DMEM containing 10% fetal bovine serum at different times.

4. Extended Data Fig. 3: could the authors clarify what is meant by “monoclonal IEC-6”? Can the authors tell whether the poly-l-lysine is really enabling the bacteria to adhere to cells or do the bacteria just stick to the poly-l-lysine substrate. We routinely use poly-l-lysine to increase the adhesion of bacteria to cells and to stick bacteria to surfaces in a non-specific manner, no images are presented to say that the bacteria are not just in locations of surface where there are not cells. Do the bacteria invade the cells following the poly-l-lysine mediated adhesion? Given the dependency of the phenotype on Collagen IV and not other collagen types (Supplementary Fig. 6) it is also a little concerning the poly-l-lysine observation may just be the result of non-specific interactions that are well known.

Response: Thanks for the critical comments and suggestions. We apologize for ignoring the detailed explanation of "monoclonal IEC-6". In order to exclude the influence of cell phenotype on the experimental results, we actually allowed a single IEC-6 cell to proliferate within a specific circular micropatterned region so that it could form a micropatterned cell monolayer. We referred to it as a monoclonal IEC-6 cell monolayer in our experiments. This has been

explained in the revised Supplementary Information. Further, we validated the effect of polylysine on the bacteria-cell adhesion. In fact, the main purpose of using polylysine was to make the IEC-6 cells adhere only to the geometrically constrained regions on the PAAm substrates. After the cell monolayers formed on the geometrically constrained micropatterns, we performed the bacterial adhesion and infection experiments. In this case, it was quite difficult for the bacteria to directly access the underlying PAAm substrates coated with poly-l-lysine because the micropatterned cell monolayers had become fully confluent. In order to clearly visualize the distribution of adhesive bacteria locations, we also performed 3D imaging of *S. aureus* infecting micropatterned cell monolayers with a laser scanning confocal microscope. It could be observed that the bacteria mainly adhere to the upper surfaces of the cell monolayers rather than the underlying PAAm substrates coated with poly-l-lysine, as displayed in the following Fig. R5 a. Moreover, we adjusted the concentration of poly-l-lysine coated on the PAAm hydrogel substrates, which showed no significant difference in bacterial adhesion at the center and edge of the micropatterned cell monolayers with spatially geometric constraints, further excluding the effect of poly-l-lysine on heterogeneous bacterial-cell adhesion (Supplementary Fig. 3e).

To investigate whether bacteria are further internalized after adhering to the cell monolayers, we characterized the internalized bacteria in the micropatterned cell monolayers infected with *S. aureus* cultured on collagen I-coating substrates and poly-l-lysine-coating substrates, respectively. Specifically, we first washed the infected micropatterned cells monolayers cultured on the substrates coated with collagen I and poly-l-lysine twice with 10 mg/ml of gentamicin to remove uninternalized *S. aureus*. Then, we lysed the cells with 0.1% Triton X-100 (Sigma-Aldrich) to release the internalized bacteria and finally perform CFUs counting. We found that there was no significantly difference in the number of bacteria internalized into the cell monolayers grown on the collagen I-coating and poly-l-lysine-coating substrates, suggesting that the bacterial adhesion and invasion are less relevant to the substances, e.g., collagen I or poly-l-lysine, used to modify the underlying substrates (Fig. R5 b-c). (See Supplementary Fig. 3a and Supplementary Fig. 3e in the Supplementary Information)

Fig. R5. Influence of PAAm hydrogel substrates coated with different substances on bacterial-cell adhesion and invasion. **a**, Representative 3D confocal images of IEC-6 cells with circular micropatterns cultured on poly-l-lysine-coating stiff hydrogel substrates infected by *S. aureus*. **b**, Representative images of internalized *S. aureus* in micropatterned cell monolayers cultured on 0.2 mg/ml collagen I-coating and 0.1 mg/ml poly-l-lysine-coating PAAm substrates, respectively. **c**, Statistical results of the number of internalized *S. aureus* in the micropatterned cell monolayers cultured on 0.2 mg/ml collagen I-coating and 0.1 mg/ml poly-l-lysine-coating PAAm substrates, respectively.

5. *Extended Data 5: could the authors indicate on the figure or in the graph what the denominator and numerator are to calculate the foldchange.*

Response: Thank you for the suggestions. In the revised version, we have added the denominator and numerator of the fold change in the figure caption of Supplementary Fig. 5a. In RNA sequencing results, the fold change essentially denotes the ratio of the FPKM (fragments per kilobase of exon model per million mapped fragments) value of the normalized gene level at the center and edge of the micropatterned cell monolayers. (See Supplementary Fig. 5a in the Supplementary Information)

6. *Extended Fig. 7: Here the authors try to conclude that the bacteria lose the ability to differentially bind to different regions of the monolayer. Aren't they really just demonstrating that collagen IV is necessary for binding. They show cells in the middle of the monolayer lack collagen IV expression, and the bulk of collagen IV present is likely below the cells in the interior of the monolayer where it would be unavailable to the bacteria for binding?*

Response: Thank you for the constructive comments and suggestions. Indeed, our experiments demonstrated that the spatially heterogeneous bacterial adhesion disappeared after the collagen IV expression in the micropatterned cell monolayers was inhibited, meaning that the collagen

IV expressed by the underlying cell monolayers played an important role in modulating heterogeneous bacteria-cell adhesion. Following the reviewer's suggestion, we further performed immunofluorescence staining of collagen IV in *S. aureus*-infected micropatterned cell monolayers to verify the necessity of collagen IV for bacterial binding. Interestingly, we found that both bacteria and collagen IV were predominantly distributed at the edges of the micropatterned cells and that many bacteria spatially colocalized with collagen IV, implying that the bacteria bind specifically to collagen IV (Fig. R6 a-b). Furthermore, we observed that the collagen IV expression was roughly homogenous throughout the cell monolayers with the aid of 3D confocal imaging, excluding the case where collagen IV was below the cell (Fig. R6 c).

To further demonstrate the specific binding of collagen IV to the bacterial collagen adhesin (Cna), we knocked out the Cna gene in *S. aureus* using the well-developed gene knockout technology. Since *S. aureus* ATCC29213 is difficult to knockout due to chloramphenicol resistance, we selected *S. aureus* USA300 that was a kind of wild bacterium (wild) to construct *S. aureus* USA300 with knockout Cna gene (Δ Cna). PCR re-validation showed that we successfully constructed this strain that knocked out the Cna gene. Next, we infected the micropatterned cell monolayers cultured on stiff substrates with *S. aureus* USA300 (wild) and *S. aureus* USA300 with knockdown Cna gene (Δ Cna), respectively. It could be found that the wild *S. aureus* could adhere to the peripheral locations of the micropatterned cell monolayers with high collagen IV expression, while the Cna-deficient *S. aureus* were not only more difficult to adhere, but also relatively evenly distribution on the underlying cell monolayers with edge constraints (Supplementary Fig. 7g-h). Likewise, we further measured the adhesion forces between the micropatterned cell monolayers and the bacteria including *S. aureus* USA300 (wild type) and *S. aureus* USA300 with knockout Cna gene (Δ Cna), respectively, which indicated that, like the previous *S. aureus* ATCC29213, the wild-type *S. aureus* USA300 showed much higher bacterial adhesion forces to the cells near the outer boundaries of the micropatterned monolayers than those at the center. Nevertheless, the adhesion forces presented by the Cna-deficient *S. aureus* were relatively low and not significantly different on the micropatterned monolayers (Supplementary Fig. 7i). These experimental findings revealed the essential contribution of collagen adhesins binding to collagen IV in the bacteria-cell interactions, and further validated that the heterogeneous expression of collagen IV caused by spatially geometric constraints on the micropatterned cell layers should be responsible for heterogeneous bacterial adhesion. Some necessary experimental results and explanations have been added to the revised

manuscript (See Page 8, Line 24-34 in the manuscript; Page 14, Line 31-42 and Page 15, Line 1-8 (Methods: Construction of *S. aureus* mutants knocking out the Cna gene) in the manuscript; Supplementary Fig. 7g-i in the Supplementary Information)

Fig. R6. Collagen IV expression in the micropatterned cell monolayers infected with *S. aureus*. **a**, Representative co-localisation images of adherent *S. aureus* and collagen IV expression in micropatterned cell monolayers. **b**, Left: the zoomed-in image of the representative co-localized image within the red window (**a**). Right: fluorescent gray values of adherent *S. aureus* and collagen IV on the micropatterned cell monolayers along the yellow dashed line (left). **c**, Representative 3D confocal images of collagen IV expression in the micropatterned cell monolayers cultured on the stiff substrates.

7. Extended Fig. 9: The fur is blocking some of the wounds, how was this controlled for? For the measurements. i.e., Were the measurements done on the animal or from the images presented such that the presence of the fur may make the measurements less reliable.

Response: In the animal experiments, we shaved the furs on the back of the rats before performing the wound infection. But, several wounds may have been concealed with the growth of fur at the later stage. Due to the relatively sparse fur coverage, we could measure the wound areas in a relatively accurate way. Besides, the Hematoxylin and Eosin (H&E) stained tissue section data could be employed to quantify the wound lengths in the absence of the furs, as presented in Fig. 5g-h, which was basically consistent with the overall trend in wound lengths.

8. Fig. 4: Monte Carlo simulation, unclear from table S2 how the beads were handled to show they result in less spatially confined binding in the spatially restricted monolayer.

Response: Thank you for your questions. We apologize for not providing a more detailed description of the bead-cell interactions in the Supplementary Information of the manuscript. In the revised version, we have added more details regarding the Monte Carlo (MC) simulation analysis. Actually, it was assumed that the forces acting on the beads (i.e., gravity, buoyancy, and Brownian random forces) in the liquid environments were the same as those exerted on the bacteria due to the fact that the densities and diameters were very close (See Supplementary Table 2 in the revised Supplementary Information of the resubmitted manuscript). So, the kinetic model of the beads was exactly the same as the bacterial kinetic one in essence if we did not consider the active interaction forces of the bacteria. It was clear that the bead-cell adhesion only involved some non-specific adhesion forces, as measured in our FluidFM-based single cell force spectroscopy (SCFS). (See Page 16 Line 29-32 in the revised Supplementary Information; Supplementary Table 2 in the revised Supplementary Information)

9. Fig. 5: The data presented in the histology are not clear. Particularly panel C, which is at such a low magnification that no bacteria can be observed to support the authors conclusion about the location and or abundance of bacteria in the tissue section. The $n=3$ is vague. Is this 3 animals or three experiments? If three experiments how many animals were used? The animal number is unclear in the methods and figure legends. The authors ignore the impact of reducing collagen on the ability of immune cells to access the tissue site and control the infection. They also ignore the possibility of greater tissue penetration by the antibiotic as alternative mechanisms rather than reduced adhesion. More concerning is what is the impact of the tyrosine kinase inhibitor on the immune responses measured in the study. This isn't to say that the inhibitor+antibiotic are more not effective than the antibiotic alone, but that it remains unclear to this reviewer whether the impact of the inhibitor is on fewer bacteria binding to the cells or a number of other factors in vivo.

Response: Thank you very much for your valuable comments and advice. In this original manuscript, all these original images were at a resolution of 300 DPI. Indeed, some images might be unclear owing to the loss of pixel resolution during the combination and processing of the images in Fig. 5. For better visualization, we have already replaced the corresponding images in Fig. 5. As such, in order to clearly show the bacteria that were stained black by Gram

stain, we have added red arrows to point out the locations of the bacteria in Fig. 5c. One could find that there were a lot of bacteria around the wounds on the first day, but very few adhered to the central areas, indicating a high probability of bacteria adhesion to the edges during wound infection. However, it appeared that the number of bacteria in the wounds decreased rapidly so that there was no significant spatial difference on the fourth day, which might be related to the regulation of autoimmunity or other potential factors. Of course, it still remains to be determined because of the complexity of the current issue involved.

In addition, we have added the detailed information about the meaning of n and the animal number in the figure legends and in the section of Methods, respectively in the revised version. In the animal experiments, we actually used a total of 30 Sprague-Dawley (SD) rats aged 60 days, each with excision wound (1 cm diameter) on their left and right backs. The rats were divided into six groups, including No infection group, PBS group, 0.1×MIC group, 1×MIC group, Inhibitor group and Inhibitor + 0.1×MIC group, in our experiments. Each group contained three duplicate samples. Specifically, n represented the number of independent experiments, i.e., excision wounds in Fig. 5.

Based on the reviewer's suggestions, we carried out immunofluorescence staining of macrophages (F4/80, BSM-34028M) and natural killer (NK) cells (CD16, bs-6028R), which are closely associated with immune responses, in the tissue sections (Yu et al. 2021; Sivori et al. 2021). We found that the expression levels of F4/80 and CD16 in the inhibitor group on the first day were not significantly different from those in the PBS group, implying that the inhibitor had a minimal effect on the immune response (Fig. R7). Likewise, the quantification of bacterial CFUs on day 1 (Fig. 5b) showed a significant reduction in infected bacteria in the inhibitor group compared to the PBS group when excluding the absence of any effect on immunity, further indicating an important role of the inhibitor *in vivo*. (See Page 11, Line 30-39 in the manuscript; Page 16, Line 11-22 (Methods: Treatment of wound infection) in the revised manuscript)

Fig. R7. a, Representative immunofluorescence images of macrophages (F4/80) and NK cells (CD16) in rat wound infections. **b,** Statistical results of the ratio of fluorescence area of macrophages (F4/80) and NK cells (CD16) in rat wound infections.

Minor concerns:

1. Dots are used in many of the graphs throughout the manuscript. They are too small to be seen.

Response: Thanks for your suggestions. In the revised manuscript, we have modified all the dots of the figures to make them more clearly presented. (See dots of all figures in the revised manuscript)

2. Line 26-27: Whereas some bacteria attach and invade cells, all bacteria do not invade cells following adherence. The sentence should be clarified.

Response: We sincerely appreciate your helpful comments and suggestions. We have narrowed the range of the bacteria that invaded the cells to avoid the incorrect statements in the revised manuscript. (See Page 1, Line 20-24 in the revised manuscript)

3. Line 40: Whereas intracellular bacteria can modify mechanotransduction, extracellular bacteria can as well. I would clarify the statement.

Response: As suggested by the reviewer, we have added the corresponding reference related to the regulation of mechanotransduction by extracellular bacteria in the revised manuscript (Petracchini et al. 2022). (See Page 1, Line 35-38 in the revised manuscript)

4. Line 43: Sca4 is produced by Rickettsia parkeri not Parkerella.

Response: This have been corrected in the revised manuscript. (See Page 2, Line 1-2 in the revised manuscript)

5. Figure 2: could you make the Y-axis labels consistent for easier comparison in panels h and i.

Response: As suggested by the reviewer, we have already corrected the Y-axis labels in the newly revised version. (See Fig. 2h-i the revised manuscript)

Reviewer #3:

In this manuscript the authors show that cells in geometrically confined environments exhibit strong spatial heterogeneity, including heterogeneous gene expression, which has a critical impact on adhesion of bacterial pathogens onto host cells. Specifically, they find that increased expression of collagen IV for cells located at edges as opposed to centers of micropatterned monolayers, leads to increased adhesion of bacteria due to enhanced adhesion forces onto the host cell surface. Moreover, this phenomenon is matrix stiffness-dependent and accentuated for cells residing on stiffer matrices. Through a Monte Carlo simulation the authors were able to recapitulate the in vitro findings in silico and, though the usage of an in vivo model they were able to show that collagen IV inhibitors increase antibiotic efficacy. Overall, I think this work is novel and interesting in that it provides new insights into how host cells interact with infectious bacteria depending on their geometrical location and how mechanical adhesion forces play a key role on those interactions. However, I have some concerns about data presentation and analysis, overstatements, and some omissions/unclear points throughout the manuscript.

Response: We sincerely thank the reviewer's positive comments and constructive suggestions, which are very helpful to revising our manuscript. In order to make it clearer, more complete, and more readable, we have accordingly supplemented some necessary experiments, statements and references in the revised manuscript, and simultaneously clarified the reviewer's concerns.

General Comments:

(1) In the Introduction there are several erroneous statements that need to be corrected. For example, authors state: "For example, the virulence factor Sca4 secreted by Parkerella specifically binds to some cell-adhesion proteins (e.g., vinculin) to reduce intercellular tension, allowing bacteria to spread more easily by manipulating cytoskeletal forces." First mistake, there is no bacterium called Parkerella. The intracellular bacterial pathogen in this study was Rickettsia parkeri. Second mistake: Sca4 does not bind to some cell-adhesion proteins. In this study it was shown that Sca4 specifically binds to vinculin, not allowing vinculin to bind to its binding partner α -catenin thus disrupting the organization of adherens junctions. Also later on the authors state: "In contrast, collective responses of epithelial monolayers to bacterial invasion can trigger the extrusion of infected cells to limit local spread of infection". This study did not look at bacterial invasion whatsoever, but rather at intercellular spread of bacteria

through host cells during the course of hours/days post-infection. Thus, this statement also needs to be corrected.

Response: Thank you for your careful review and professional suggestions. In the latest manuscript, we have corrected the word of “*Parkerella*” into “*Rickettsia parkeri*” and more accurately described the specific binding of Sca4 to vinculin. Additionally, we have updated the inappropriate citation and simultaneously added a more relevant literature to support the viewpoint that “the collective response of the epithelial monomolecular layer to bacterial invasion can trigger extrusion of infected cells to limit the local spread of infection” (Bastounis et al. 2022; Bastounis et al. 2021). (See Page 2, Line 1-4 in the revised manuscript)

(2) In the last paragraph of the introduction the authors state: “With the aid of well-established microcontact printing technique, we first develop an in vitro model of bacteria infecting cells, which allows pathogenic bacteria to infect host cell monolayers grown on spatially confined extracellular substrates. We demonstrate that there are remarkably spatially heterogeneous bacteria-host interactions owing to the presence of two-dimensional (2D) geometric constraints on host monolayers.” The authors only tested two different bacterial pathogens, namely S. aureus and E. coli, infecting just two cell lines, namely HACAT and IEC6. Statements like the above, sound like overstatements and misleading. It looks like the authors findings apply to all cell lines and bacterial pathogens, which is not the case or at least has not been tested. So, if they want to keep the general statements, they would have to test a lot more pathogens and see if same findings apply. Otherwise, I would introduce from the introduction the model pathogens tested and give some background as to why those were chosen to be tested. Along that line it looks like “Collagen IV” is an important receptor for adhesion of E. coli and S. aureus and that would be nice to be introduced somehow, since it is something already known. Also, is it an important adhesion receptor for more bacterial pathogens? Finally, the statement in the discussion “indicating that the absolute magnitude of bacterial adhesion forces were largely dependent on the specific type of bacteria” it is clearly an overstatement since only two bacterial pathogens were tested. This has to either be more specific or experiments with more bacterial pathogens have to be conducted.

Response: Thank you for your critical comments and instructive suggestions. Following your suggestions, we have provided a more detailed description of the pathogenic bacteria and cells we used in the work to avoid possible misunderstandings in the subsequent conclusions. As

seen in the manuscript, we chosen two typical bacterial pathogens, i.e., Gram-positive bacteria (*S. aureus*) and Gram-negative bacteria (*E. coli*), to infect two kinds of epithelial cells (i.e., IEC-6 cells and HaCat cells) that were frequently adopted in the bacteria-cell interactions (Ribet and Cossart 2015). It was found there existed significant spatial heterogeneity in the interactions between these two bacteria and the host epithelial cells due to the presence of two-dimensional (2D) geometric constraints on the host cell monolayers. Furthermore, we have added some necessary statements that “In addition to being an important adhesion receptor for *S. aureus* and *E. coli*, collagen IV can also adhere to various bacterial pathogens, e.g., *Streptococcus pneumoniae*, *Helicobacter pylori*, and *E. faecalis*” as reported in some literatures (Kostrzynska and Wadstrom 1992; Trust et al. 1991; Nallapareddy et al. 2000). Besides, we have already corrected our statements about the bacterial types in the discussion by directly pointing out the two bacterial pathogens used in the experiments (*S. aureus* and *E. coli*), and also minimized the degree of correlation between the adhesion forces and the bacterial types to avoid the potential overstatements as much as possible. (See Page 2, Line 12-16 in the revised manuscript; Page 6, Line 37-42 in the revised manuscript; Page 13, Line 18-21 in the revised manuscript)

(3) A general very important comment: Could it just be the cell shape and not the geometric constraint per se that drives the differential adhesion of bacteria onto host cells? I am wondering how much the geometric constraints are responsible for the spatially heterogeneous bacteria-cell adhesion pattern as compared to the cell shape. Can the authors use the F-actin signal (since it looks in Figure 1 that it localizes at the cell-cell edges) or some cell-cell edge marker (e.g., anti-E-cadherin antibody) to segment cells and figure out how the shape of cells at the edges differs from the shape of cells at the center? Could it be that the distinct shape of the cells is what governs changes in gene expression and thus bacterial pathogen adhesion onto host cells? Could it be that on soft matrices cells at the edges are more similar to cells at the center (as opposed to stiff) and that is why the heterogeneity of bacterial adhesion is decreased? In line with that if one looks at the images in Figure 1D, it looks like on the soft hydrogel the cells are more homogenously spread, or that cells at edges look more like cells in the middle. This is not the case on stiff hydrogel. So, is there a way of assessing the role of cell shape on modulating bacterial adhesion (potentially independent of geometric constraint)?

Response: Thank you very much for your valuable comments. According to the reviewer's suggestions, we have further quantified some indices associated with cell shape, such as

circularity of single cells that was defined as $\text{circularity} = 4\pi \times \text{area}/\text{perimeter}^2$ (Zhao et al. 2019). To this end, we first segmented the cells in the micropatterned monolayers based upon their F-actin and DAPI fluorescence signals to obtain the area and perimeter of individual cells via the well-developed Cellprofiler software, as shown in the following Fig. R8. The corresponding statistical results showed that cell circularity at the central regions of the micropatterned cell monolayers was not significantly different from that at the edges, regardless of soft, medium or stiff PAAm substrates. Likewise, we found that there was no significant difference in the circularities of the cells cultured on the PAAm substrates with different rigidities, although their mean values slightly decreased with increasing the substrate stiffness, as displayed in Fig. R8. Further, it was found that the circularity of the cells under spatially constraints was not significantly different from that of the cells without constraints. Additionally, we performed the bacteria-cell adhesion experiments based on IEC-6 cell monolayers without prescribed geometric constraints, as shown in the following Fig. R3. Specifically, one could observe that the bacterial adhesion was spatially random whereas the geometrical shapes of the IEC-6 cells were different from each other in essence, as presented in the following Fig. R3. All these facts suggest that the bacterial-cell adhesion phenomenon should be less affected by the cell shapes. In the revised version, we have added some necessary experimental results as mentioned above. (See Page 4, Line 9-15 in the revised manuscript; Supplementary Fig. 1g in the revised Supplementary Information)

Fig. R8. Top: representative segmentation images of individual cells in the micropatterned cell monolayers cultured on the PAAm substrates with different stiffnesses. Bottom: the corresponding statistical results of cell circularity at the center and at the edge of these micropatterned cell monolayers.

Fig. R3. Representative images of high-density IEC-6 cell monolayers without geometric constraints infected by *S. aureus* expressing GFP.

(4) In the legend of Figure 1 it is stated: “All these measured data were shown as mean \pm SD ($n=5$; two-tailed unpaired t -test), unless otherwise stated.” What does $n=5$ mean? Five independent hydrogels? Or five independent islands within the same hydrogel? I think if it is five independent islands (i.e., just five images) definitely more images need to be taken on hydrogels prepared on independent days, since the variability in stiffness of polyacrylamide hydrogels is well documented. The same question applies to figures shown later on (e.g. Figure 3). Also, for comparing distributions authors should probably better use a non-parametric test rather than a t -test (like Wilcoxon ranksum). Also, why did the authors chose 10 kPa, 32 kPa and 93 kPa? Is there any (patho)physiological relevant behind those chose values of matrix stiffness? This should be explained.

Response: Thank you for your useful suggestions. In the revised manuscript, we have given the number of independent samples adopted in the experiments and the specific meaning of n in the figure. Also, we have already improved the method of statistical analysis based on the attributes of these data samples involved. Briefly, we applied Wilcoxon ranksum test for two data groups that did not satisfy normality and the homogeneity of variances, two-tailed unpaired t -tests for two data groups that approximately met normality and the homogeneity of variances and one-way ANOVAs for more than three data groups. In addition, we selected the PAAm substrates whose stiffness was 10 kPa, 32 kPa and 93 kPa, respectively, which were in the range of normal tissue stiffness *in vivo* (Janmey et al. 2020). It could ensure that the microenvironments of bacteria-cell interactions were closer to that *in vivo*. In practice, we prepared the PAAm substrates according to the formulation described in some previous studies (Denisin and Pruitt 2016), and their Young’s moduli were measured through a Piuma nanoindenter equipped with a Chiaro indenter head (Optics11 Life, Netherlands, as shown in

the following Fig. R9), as described in the section of “**Characterization of PAAm hydrogel stiffness**” in the Supplementary Information of the revised manuscript. (See Page 4, Line 6-8 in the revised manuscript; Page 14, Line 1-12 in the Supplementary Information; Supplementary Fig. 1e-f in the Supplementary Information)

Fig. R9. Representative load-displacement curves of PAAm substrate with different rigidities (soft, medium and stiff) based on a nanoindenter. **f**, Elastic modulus (E) of PAAm substrate with different rigidities (soft, medium and stiff).

(5) *Authors state: “To further assess the generality of our findings, we changed the geometric constraint parameters of micropatterned cell monolayers, including their size (100, 200 and 400 μm diameter). Under these conditions, the spatially heterogeneous adhesion phenomenon remained unchanged, indicating that the bacteria – host adhesion behavior was independent of the size of geometric constraints” . This is a bit confusing because if one looks at Figure 1c it is clear that the larger the diameter, the bigger the difference in adhesion between center and edge cells. If one compares $D=100$ μm versus $D=400$ μm, the relative difference in the latter case ($D=400$) is at least four times bigger than in the former ($D=100$). So I think it is not fair to claim that the diameter does not play a role (when it does) but then say that the matrix stiffness plays a role when the relative increase on stiff matrix as compared to soft is approximately three-fold if one looks at the boxplots of Figure 1f. Back again to the comment #3 I made, I speculate that the shape of cells at edges versus center might play a key role on the observations made above.*

Response: Thank you for your valuable suggestions. As pointed out by the reviewer, the fluorescence intensities of bacteria adhered to the edge areas of the large-diameter micropatterned monolayers were indeed higher than those of bacteria attached to the edge regions of the small-diameter monolayers. In this work, our main goal was to show that the

spatially heterogeneous adhesion phenomenon still occurred as the size of the micropatterned monolayers changed, so didn't directly compare the number of bacteria adhered to these micropatterned monolayers with different sizes. In the revised manuscript, we have removed the conclusive statement that "bacteria-host adhesion behavior is not related to the size of the geometric constraints and added the generality of heterogeneous adhesion across the different sizes of geometric constraint". Furthermore, we have also calculated the circularity of individual cells in cell monolayers of different geometrically constrained sizes (D=100 and 400 μm in diameter) to further explore the effect of single cell shape (Fig. R10). The corresponding quantitative data indicated no significant difference in the circularities of the cells located near the central and peripheral regions for the micropatterned cell monolayers with D=100 and 400 μm , which implied that the bacteria-cell heterogeneous adhesion interactions were less affected by the cell shapes. (See Page 4, Line 3-6 in the revised manuscript)

Fig. R10. **a**, Representative segmentation images of individual cell shapes in the IEC-6 cell monolayers with micropatterns of different sizes (D=100 and 400 μm in diameter, respectively). **b**, Statistical results of cell circularities at the center and at the edge of the cell monolayers with D=100 and 400 μm .

*(6) Do higher adhesion forces and higher adhesion suggest higher host cell invasion? Previously fit was shown *Listeria monocytogenes* adheres more on endothelial cells residing on stiffer extracellular matrices as opposed to softer, and that ultimately to more bacterial invasion (1)? I think this experiment might be very interesting to perform and relatively easy with some differential immunostaining of the previously fixed samples or with confocal imaging of the current ones.*

Response: According to your suggestions, we have further explored bacterial invasion between *S. aureus* and the IEC-6 cell monolayers cultured on soft, medium and stiff substrates, respectively. Due to the lack of available anti-*S. aureus* antibodies in the laboratory, it was difficult for us to directly distinguish whether it is an adherent but not internalized bacterium

or a fully internalized bacterium by antibody-specific labelling methods, as you mentioned in some previous studies (Bastounis et al. 2018). To this end, we treated infected cells with 10 mg/ml gentamicin to remove the uninternalized bacteria without disturbing the internalized cells (Schirmer et al. 2016). Then, we detected the cellular infection rate of internalized bacteria based on a flow cytometry and the number of internalized bacteria with the help of laser confocal imaging and CFU counting methods, as shown in the following Fig. R11. Interestingly, it was found that, in contrast to *Listeria monocytogenes* that was highly invasive to endothelial cells cultured on stiff extracellular matrix (Bastounis et al. 2018), *S. aureus* exhibited relatively low bacterial invasion of IEC-6 cell monolayers grown on the stiff substrates (Liu et al. 2021). This suggests that there may be differences in bacterial adhesion and invasion between different bacterial types and cell types (Fig. R11).

Fig. R11. Dependence of invasion of *S. aureus* to IEC-6 cell monolayers on substrate rigidities. a-b, Representative images (a) and statistical results (b) of cellular infection rate of internalized bacteria on

substrates with different rigidities based on the flow cytometry. **c-d**, Representative confocal images (**c**) and statistical results (**d**) of the number of bacteria internalized to IEC-6 cell monolayers cultured on substrates with different rigidities. **e-f**, Representative images (**e**) and statistical results (**f**) of the number of bacteria internalized to IEC-6 cell monolayers cultured on substrates with different rigidities based on the colony-count technique.

Minor comments:

(1) In figure 4 authors show how matrix stiffness impacts the expression of collagen IV. What about if one was to compare $D=100$ to $D=400$ μm islands? Is the difference going to be smaller for $D=100$ as expected?

Response: Thank you for your suggestions. By means of immunofluorescence staining of collagen IV expressed by the circular micropatterned monolayers whose diameters were $D=100$ and 400 μm , respectively, we found that the expression of collagen IV at the edge regions was much higher than that at the central regions regardless of the diameters of the micropatterned monolayers (see the following Fig. R12 for details). Interestingly, it turned out that the average fluorescence intensities slightly increased when the diameter of the micropatterned cell monolayers increased from $D=100$ to 400 μm , as seen in the following Fig. R12.

Fig. R12. Immunofluorescence staining results (cyan) of collagen IV expressed in the cell monolayers with different diameters (100 and 400 μm), and the statistical results of normalized fluorescence intensities at the center and at the edge of the micropatterned cell monolayers.

(2) Because of color blind people use or red and green (especially overlay) should be avoided. For example, see immunofluorescence images in Figure 1. Maybe change the red color to magenta?

Response: According to the reviewer's suggestions, we have already replaced red with magenta in all the red-green overlaid immunofluorescence images for better presentation to all readers (See all these images involved in the revised manuscript). Since the cyan, magenta and blue overlaid images (without green) were similar in color, we did not change red in this case (See two typical images for comparison in the following Fig. R13).

Fig. R13. A comparison of cyan, red and blue overlaid image with cyan, magenta and blue overlaid one.

(3) In the boxplots used in Figure 2 h, Figure 3 g etc it would be nice to add the individual data points pretty much as authors do for Figure 1 at the barplots. That would increase transparency and give a sense of the variability between experimnts.

Response: According to your useful suggestion, we have added the individual data points for all box and bar charts similar to Fig. 1 and also provided the corresponding source data. (See box and bar charts of all figures in the revised manuscript)

(4) In Figure 2G why are beads only and *S. aureus* shown and not *E. coli*?

Response: Thank you for your suggestion. Since most experiments were performed with *S. aureus* as a representative infection, we focused on comparing the adhesion forces between *S. aureus* and the beads on the micropatterned cell monolayers. Based on your suggestions, we have further supplemented the adhesion forces between *E. coli* and the underlying micropatterned cell monolayers at different locations, which increased with radius, as shown in the following Fig. R14.

Fig. R14. The adhesion forces for *E. coli* along the radial direction of the circular cell monolayers.

References

- 1 Sarangi, B. R. *et al.* Coordination between intra- and extracellular forces regulates focal adhesion dynamics. *Nano Lett.* **17**, 399-406 (2017).
- 2 Bastounis, E. E., Radhakrishnan, P., Prinz, C. K. & Theriot, J. A. Mechanical forces govern interactions of host cells with intracellular bacterial pathogens. *Microbiol. Mol. Biol. Rev.* **86**, e0009420 (2022).
- 3 Liu, X. *et al.* Extracellular matrix stiffness modulates host-bacteria interactions and antibiotic therapy of bacterial internalization. *Biomaterials* **277**, 121098 (2021).
- 4 Elosegui-Artola, A. *et al.* Mechanical regulation of a molecular clutch defines force transmission and transduction in response to matrix rigidity. *Nat. Cell Biol.* **18**, 540 (2016).
- 5 Chen, S. Y., Lin, J. S. & Yang, B. C. Modulation of tumor cell stiffness and migration by type IV collagen through direct activation of integrin signaling pathway. *Arch. Biochem. Biophys.* **555**, 1-8 (2014).
- 6 Helenius, J., Heisenberg, C. P., Gaub, H. E. & Muller, D. J. Single-cell force spectroscopy. *J. Cell. Sci.* **121**, 1785-1791 (2008).
- 7 Janmey, P. A., Fletcher, D. A. & Reinhart-King, C. A. Stiffness sensing by cells. *Physiol. Rev.* **100**, 695-724 (2020).
- 8 Denisin, A. K. & Pruitt, B. L. Tuning the range of polyacrylamide gel stiffness for mechanobiology applications. *ACS Appl. Mater. Interfaces* **8**, 21893-21902 (2016).
- 9 Bastounis, E. E., Yeh, Y. T. & Theriot, J. A. Matrix stiffness modulates infection of endothelial cells by *Listeria monocytogenes* via expression of cell surface vimentin. *Mol. Biol. Cell* **29**, 1571-1589 (2018).
- 10 de Castro, L. F. *et al.* Secreted frizzled related-protein 2 (Sfrp2) deficiency decreases adult skeletal stem cell function in mice. *Bone Res.* **9**, 49 (2021).
- 11 Wang, D. *et al.* FAM83D activates the MEK/ERK signaling pathway and promotes cell proliferation in hepatocellular carcinoma. *Biochem Bioph Res Co* **458**, 313-320 (2015).
- 12 Dyer, D. P. *et al.* TSG-6 Inhibits neutrophil migration via direct interaction with the chemokine CXCL8. *J. Immunol.* **192**, 2177-2185 (2014).
- 13 Schutyser, E. *et al.* Regulated production and molecular diversity of human liver and activation-regulated chemokine/macrophage inflammatory protein-3 alpha from normal and transformed cells. *J. Immunol.* **165**, 4470-4477 (2000).
- 14 Hamidouche, Z. *et al.* Priming integrin alpha 5 promotes human mesenchymal stromal cell osteoblast differentiation and osteogenesis. *Proc. Natl. Acad. Sci. U.S.A.* **106**, 18587-18591 (2009).
- 15 Sepulveda, J. L. & Wu, C. The parvins. *Cell. Mol. Life Sci.* **63**, 25-35 (2006).
- 16 Otsubo, C. *et al.* TSPAN2 is involved in cell invasion and motility during lung cancer progression. *Cell Rep.* **7**, 527-538 (2014).

- 17 Hong, S. N. *et al.* Duox2 is required for the transcription of pattern recognition receptors in acute viral lung infection: An interferon-independent regulatory mechanism. *Antiviral Res.* **134**, 1-5 (2016).
- 18 Wang, X. H. *et al.* Comprehensive analysis of TP53 and CD58 mutations and identification of patients with inferior prognosis and enhanced immune escape in diffuse large B cell lymphoma. *Blood* **140**, 9236-9237 (2022).
- 19 Zhang, M. X. *et al.* USP20 promotes cellular antiviral responses via deconjugating K48-Linked ubiquitination of MITA. *J. Immunol.* **202**, 2397-2406 (2019).
- 20 Yu, J. L. *et al.* Liver metastasis restrains immunotherapy efficacy via macrophage-mediated T cell elimination. *Nat. Med.* **27**, 152 (2021).
- 21 Sivori, S. *et al.* NK cells and ILCs in tumor immunotherapy. *Mol. Aspects Med.* **80**, 100870 (2021).
- 22 Petracchini, S. *et al.* Optineurin links Hace1-dependent Rac ubiquitylation to integrin-mediated mechanotransduction to control bacterial invasion and cell division. *Nat. Commun.* **13**, 6059 (2022).
- 23 Bastounis, E. *et al.* Mechanical competition triggered by innate immune signaling drives the collective extrusion of bacterially infected epithelial cells. *Dev. Cell* **56**, 443-460 (2021).
- 24 Ribet, D. & Cossart, P. How bacterial pathogens colonize their hosts and invade deeper tissues. *Microbes Infect.* **17**, 173-183 (2015).
- 25 Kostrzynska, M. & Wadstrom, T. Binding of laminin, type-IV collagen, and vitronectin by Streptococcus-Pneumoniae. *Zentralbl Bakteriolog.* **277**, 80-83 (1992).
- 26 Trust, T. J. *et al.* High-affinity binding of the basement-membrane proteins collagen type-IV and laminin to the gastric pathogen Helicobacter-Pylori. *Infect. Immun.* **59**, 4398-4404 (1991).
- 27 Nallapareddy, S. R., Qin, X., Weinstock, G. M., Hook, M. & Murray, B. E. Enterococcus faecalis adhesin, ace, mediates attachment to extracellular matrix proteins collagen type IV and laminin as well as collagen type I. *Infect. Immun.* **68**, 5218-5224 (2000).
- 28 Zhao, Y., Sun, Q., Wang, S. R. & Huo, B. Spreading shape and area regulate the osteogenesis of mesenchymal stem cells. *Tissue Eng Regen Med* **16**, 573-583 (2019).
- 29 Schirmer, M. *et al.* Linking the human gut microbiome to inflammatory cytokine production capacity. *Cell* **167**, 1125 (2016).

REVIEWER COMMENTS

Reviewer #1 (Remarks to the Author):

I am happy with the revision and with all answers provided

Reviewer #2 (Remarks to the Author):

The authors were highly responsive to the initially critiques and have adequately addressed my concerns.

Reviewer #3 (Remarks to the Author):

Manuscript Summary:

In this manuscript the authors show that cells in geometrically confined environments exhibit strong spatial heterogeneity, including heterogeneous gene expression, which critically impacts adhesion of bacterial pathogens onto host cells. Increased expression of collagen IV for host cells located at edges as opposed to centers of micropatterned monolayers, leads to increased bacterial adhesion due to enhanced adhesion forces onto the host cell surface. Moreover, this phenomenon is matrix stiffness-dependent and accentuated for cells residing on stiffer matrices. Overall, I think this work is novel and interesting, and authors have addressed quite of my prior concerns. Below please find some further comments I have that I think are important to be addressed.

Comments:

(1) In previous comment #3 that I made, I raised the question of whether cell shape and not geometric constraints per se drives the differential adhesion of bacteria into host cells. This in turn could be due to for example to differential gene expression patterns, or amounts of host cell surface receptors present or accessible etc. I was triggered to say so, because it looked that the area of cells at edges vs center was more similar when those cells resided on soft as opposed to stiff substrates. The authors went ahead and claimed they calculated some indices associated with cell shape, but in reality they only calculated the "circularity" of cells, that is how elongated or conversely circular cells are and found no significant differences. However, circularity is just one shape parameter that cells could modulate, which neither by eye nor by quantitation seems to differ between conditions. What about other shape parameters, most importantly the cell area? Given that cell segmentation has already been performed by authors, it should be straightforward to examine whether the cell area varies between edges vs center, or based on matrix stiffness.

(2) The authors also claim that there were no significant differences in the circularities of cells grown without constraints as compared to cells grown with constraints. But where is the quantitation for that? Was cell segmentation performed in this case, and where is the corresponding barplot? If not, coming into such conclusion just by looking at immunofluorescence images does not look very convincing.

(3) In previous comment #4 I made, I mentioned that authors should comment on the choice of the particular substrate stiffnesses they chose, namely 10 kPa, 32 kPa and 93 kPa and discuss whether there is any (patho)physiological relevance behind those choices. Authors responded to my previous comment by saying: "In addition, we selected the PAAm substrates whose stiffness was 10 kPa, 32 kPa and 93 kPa, respectively, which were in the range of normal tissue stiffness in vivo (Janmey et al. 2020). It could ensure that the microenvironments of bacteria-cell interactions were closer to that in vivo." Close to which tissue stiffness? Brain tissue is very soft and of the

order of 1 kPa or less, while bone tissue is orders of magnitude stiffer. Authors reference a review paper (Janmey, P. A et al. *Physiol. Rev.* 100, 695-724, 2020) but I do not see there how the three stiffnesses the authors chose are somehow mentioned there. Authors examine two host epithelial cell types, IEC-6 which are intestinal epithelial cells and HaCat which are epithelial cells from human skin. Do the values of stiffness they chose somehow relate to ECM stiffness of healthy skin or intestine? In fibrotic diseases for example are there any measurements reported showing how much stiffer those particular ECMs can become?

(4) In response to previous comment #6 I made, authors went ahead to examine whether higher bacterial adhesion leads to higher host cell invasion and how that depends on matrix stiffness, using flow cytometry. They found that on what they define as "stiff matrix" (namely 93 kPa) there was significantly more invasion as compared to what they define as "soft matrix" (namely 10 kPa). In Fig. R11b where they show the results of this experiment using barplots, it appears that there are only three points. Is that one experiment only? Are those points coming from measurements done on three different infected wells? Moreover, what test was used to assess whether there is a significant difference? Nonparametric test should be used unless the distributions are normal. Finally on soft matrices approx. 35% of cells get infected while on stiff approx. 30%. Therefore, the decrease in bacterial invasion is just 14%. On the contrary in the study involving infection of endothelial cells by *Listeria monocytogenes* the difference in invasion between stiff and soft was approximately two-fold. Moreover, in that particular study soft matrices considered were 0.6 kPa and 3 kPa. Therefore, one should be cautious when comparing different studies involving different ECM stiffnesses (i.e., "soft" and "stiff" conditions are different in this case), different host cell types and pathogens. I.e., the discrepancy between these two studies could be due to the fact that in the present study no experiments were done on matrices below 10 kPa, or it could be the host cell type or the specific pathogens.

Reviewer #4 (Remarks to the Author):

Comments on MC simulations for the paper "Spatially heterogeneous bacteria-host adhesion dominated by geometric constraint-triggered collagen IV expression with edge effects"

The paper and the Supplementary Information file give sufficient information on the MC simulations of the bacterial dynamics. Overall, I would propose the publication of the manuscript. However, there are some questions regarding the method:

1. What is the ratio of the two-dimensional square plane used in the simulations with regard to real bacterial dimensions? Does each square in the lattice represent an area of a few micron where the bacteria is either present or absent? What is the size of the grid with boundary periodic conditions used and what is the rationale behind using such scale?
2. How does gravity play into the bacteria dynamics if it takes place only on a plane (50 μm above the substrate on a monolayer)? It seems that the bacteria can move only laterally and not vertically. This implies that omission of the gravity term in equation (2) of Supplementary Information would yield similar results.
3. What condition is satisfied and with what probability when bacterium receptor-ligand binds and dissociates?

Point-by-point responses to the reviewers' comments and suggestions

(Manuscript Number: NCOMMS-23-11807B)

Reviewer #1:

I am happy with the revision and with all answers provided.

Response: We sincerely thank the reviewer for his/her positive comments and constructive suggestions, which were very helpful for improving the quality of our manuscript.

Reviewer #2:

The authors were highly responsive to the initially critiques and have adequately addressed my concerns.

Response: We sincerely thank the reviewer for his/her positive comments and constructive suggestions, which were very helpful for improving the quality of our manuscript.

Reviewer #3:

In this manuscript the authors show that cells in geometrically confined environments exhibit strong spatial heterogeneity, including heterogeneous gene expression, which critically impacts adhesion of bacterial pathogens onto host cells. Increased expression of collagen IV for host cells located at edges as opposed to centers of micropatterned monolayers, leads to increased bacterial adhesion due to enhanced adhesion forces onto the host cell surface. Moreover, this phenomenon is matrix stiffness-dependent and accentuated for cells residing on stiffer matrices. Overall, I think this work is novel and interesting, and authors have addressed quite of my prior concerns. Below please find some further comments I have that I think are important to be addressed.

Response: We appreciate the reviewer's positive comments and constructive suggestions, which are very helpful to revising our manuscript. Accordingly, we have added some necessary discussion and references in the revised manuscript to further clarify the reviewers' concerns.

General Comments:

(1) In previous comment #3 that I made, I raised the question of whether cell shape and not geometric constraints per se drives the differential adhesion of bacteria into host cells. This in turn could be due to for example to differential gene expression patterns, or amounts of host cell surface receptors present or accessible etc. I was triggered to say so, because it looked that the area of cells at edges vs center was more similar when those cells resided on soft as opposed to stiff substrates. The authors went ahead and claimed they calculated some indices associated with cell shape, but in reality they only calculated the “circularity” of cells, that is how elongated or conversely circular cells are and found no significant differences. However, circularity is just one shape parameter that cells could modulate, which neither by eye nor by quantitation seems to differ between conditions. What about other shape parameters, most importantly the cell area? Given that cell segmentation has already been performed by authors, it should be straightforward to examine whether the cell area varies between edges vs center, or based on matrix stiffness.

Response: Thank you so much for your instructive comments and suggestions. Following your suggestions, we have further quantified the areas of individual cells in the micropatterned cell monolayers cultured on the substrates with different rigidities through cell segmentation based on the Cellprofiler software. The corresponding statistical results indicated that there were no significant differences in the cell areas between the central and peripheral regions of the micropatterned cell monolayers cultured on the soft, medium and stiff substrates, as shown in the following **Fig. R1**. This has been further emphasized in the newly revised manuscript (See Page 4, Line 9-13 in the revised manuscript and Supplementary Table 1).

Fig. R1. Statistical results of cell area at the center and edge of the micropatterned cell monolayers cultured on soft, medium and stiff substrates.

Although the cell areas in the center and edge of micropatterned cell monolayers on soft substrates might be closer to each other relative to those on medium and stiff substrates, bacteria

still tended to adhere to the edges of the cell monolayers, regardless of the underlying substrate stiffness. In the absence of geometric constraints, we observed that bacterial adhesion was spatially random, whereas the geometrical shapes of the IEC-6 cells were qualitatively different based on their cell circularity (See Supplementary Fig. 1a and Supplementary Fig. 1g). These results further suggest that the cell shape may not be a major factor responsible for the spatially heterogeneous bacterial adhesion (See Supplementary Fig. 1g).

(2) The authors also claim that there were no significant differences in the circularities of cells grown without constraints as compared to cells grown with constraints. But where is the quantitation for that? Was cell segmentation performed in this case, and where is the corresponding barplot? If not, coming into such conclusion just by looking at immunofluorescence images does not look very convincing.

Response: Thank you very much for your professional comments. We apologize for neglecting the cell circularity data for cell monolayers without geometric constraints. In the latest revision, we have segmented the cell monolayers without geometric constraints to quantify their cell circularity, which indicated that there was no significant difference in the cell circularity between the cells cultured on substrates with and without geometric constraints (See Page 4, Line 13-16 in the newly revised manuscript and Supplementary Fig. 1g).

(3) In previous comment #4 I made, I mentioned that authors should comment on the choice of the particular substrate stiffnesses they chose, namely 10 kPa, 32 kPa and 93 kPa and discuss whether there is any (patho)physiological relevance behind those choices. Authors responded to my previous comment by saying: “In addition, we selected the PAAm substrates whose stiffness was 10 kPa, 32 kPa and 93 kPa, respectively, which were in the range of normal tissue stiffness in vivo (Janmey et al. 2020). It could ensure that the microenvironments of bacteria-cell interactions were closer to that in vivo.” Close to which tissue stiffness? Brain tissue is very soft and of the order of 1 kPa or less, while bone tissue is orders of magnitude stiffer. Authors reference a review paper (Janmey, P. A et al. Physiol. Rev. 100, 695-724, 2020) but I do not see there how the three stiffnesses the authors chose are somehow mentioned there. Authors examine two host epithelial cell types, IEC-6 which are intestinal epithelial cells and HaCat which are epithelial cells from human skin. Do the values of stiffness they chose somehow relate to ECM stiffness of healthy skin or intestine? In fibrotic diseases for example

are there any measurements reported showing how much stiffer those particular ECMs can become?

Response: Thanks for your further comments and suggestions. In the newly revised version, we have added more descriptions and references to clarify the reason why we adopted the specific substrate rigidities in the current work. As stated by the reviewer, the stiffness of living tissues usually ranges from Pascals to Gigapascals. Generally, the Young's moduli of the intestines and skins are ~1-40 kPa and ~2-850 kPa, respectively (Guimaraes et al. 2020; Handorf et al. 2015; He et al. 2023). However, in certain pathological conditions such as fibrosis and cancer, tissue stiffening is likely to occur (Guimaraes et al. 2020). For example, the Young's modulus of healthy colon tissues is ~1 kPa whereas it increases by at least 1 order of magnitude in the colon tissue with fibrotic inflammatory bowel disease (He et al. 2023). The skin stiffness of patients with systemic sclerosis is ~47.32 kPa, which is more than twice that of normal subjects (~19.5 kPa) (Tumsatan et al. 2022). Since intestinal epithelial cells and keratinocytes were adopted in the current work, we therefore selected the PAAm hydrogel substrates whose Young's moduli were 10.14, 32.29 and 93.46 kPa, respectively, to mimic mechanical microenvironments of the above-mentioned cells in physiological and pathological conditions. This has been further emphasized in the newly revised manuscript (See Page 4, Line 4-6 in the revised manuscript).

*(4) In response to previous comment #6 I made, authors went ahead to examine whether higher bacterial adhesion leads to higher host cell invasion and how that depends on matrix stiffness, using flow cytometry. They found that on what they define as “stiff matrix” (namely 93 kPa) there was significantly more invasion as compared to what they define as “soft matrix” (namely 10 kPa). In Fig. R11b where they show the results of this experiment using barplots, it appears that there are only three points. Is that one experiment only? Are those points coming from measurements done on three different infected wells? Moreover, what test was used to assess whether there is a significant difference? Nonparametric test should be used unless the distributions are normal. Finally on soft matrices approx. 35% of cells get infected while on stiff approx. 30%. Therefore, the decrease in bacterial invasion is just 14%. On the contrary in the study involving infection of endothelial cells by *Listeria monocytogenes* the difference in invasion between stiff and soft was approximately two-fold. Moreover, in that particular study soft matrices considered were 0.6 kPa and 3 kPa. Therefore, one should be cautious when*

comparing different studies involving different ECM stiffnesses (i.e., “soft ” and “stiff” conditions are different in this case), different host cell types and pathogens. I.e., the discrepancy between these two studies could be due to the fact that in the present study no experiments were done on matrices below 10 kPa, or it could be the host cell type or the specific pathogens.

Response: Thanks for your instructive comments and suggestions. For Fig. R11b in our previous revision, we all performed three different independent infection experiments and compared the invasion rates of the three groups with different substrate rigidities using one-way ANOVAS. We found that *S. aureus* exhibited relatively low bacterial invasion of IEC-6 cell monolayers grown on the stiff substrates (93.46 kPa) (Liu et al. 2021), while *Listeria monocytogenes* were highly invasive to endothelial cells cultured on stiff extracellular matrix (Bastounis et al. 2018). As mentioned by the reviewer, several factors like ECM stiffness, host cell types, and pathogen types, may affect bacterial invasion. To this end, we have already added some necessary descriptions and references in the discussion section (See Page 13, Line 18-23 in the revised manuscript; Supplementary Fig. 11 in the Supplementary Information; Page 18, Line 5-20 in the Supplementary Information).

Reviewer #4:

Comments on MC simulations for the paper “Spatially heterogeneous bacteria – host adhesion dominated by geometric constraint-triggered collagen IV expression with edge effects ”
The paper and the Supplementary Information file give sufficient information on the MC simulations of the bacterial dynamics. Overall, I would propose the publication of the manuscript. However, there are some questions regarding the method:

Response: We sincerely thank the reviewer’s positive comments and constructive suggestions, which are very helpful for revising our manuscript. Below, we have clarified the reviewer’s concerns and indicated the corresponding revisions made in the latest revised manuscript and Supplementary Information.

1. What is the ratio of the two-dimensional square plane used in the simulations with regard to real bacterial dimensions? Does each square in the lattice represent an area of a few micron where the bacteria is either present or absent? What is the size of the grid with boundary periodic conditions used and what is the rationale behind using such scale?

Response: Thank you for your critical comments and useful suggestions. In the newly revised manuscript, we have supplemented the related contents including mesh size of the two-dimensional square plane, size of periodic boundary conditions and the rationale behind using such scale in the present Monte Carlo simulations, as pointed out by the reviewer. As a matter of fact, considering that the diameter of the bacteria was on the order of $\sim 1 \mu\text{m}$, we set the side length of the square lattice to be $1 \mu\text{m}$ to ensure that a single bacterium could completely adhere to a specific square lattice (See Supplementary Table 3 for details). Further, the sizes of the two-dimensional square planes were designed as 300, 400, and 600 μm , to simulate the micropatterned cell monolayers of 100, 200 and 400 μm in diameter, respectively, as illustrated in Supplementary Fig. 9 in the Supplementary Information. Additionally, we imposed periodic boundary conditions at the boundaries of cubic simulation regions above the two-dimensional square planes to ensure that the total number of bacteria was constant during the Monte Carlo simulations (See Supplementary Table 3, Supplementary Fig. 9 and Page 16, Line 12-25 in the Supplementary Information).

2. How does gravity play into the bacteria dynamics if it takes place only on a plane (50 μm above the substrate on a monolayer)? It seems that the bacteria can move only laterally and not vertically. This implies that omission of the gravity term in equation (2) of Supplementary Information would yield similar results.

Response: Thanks for your instructive comments and suggestions. Indeed, the bacteria moved randomly in a liquid environment. To improve the efficiency of the numerical simulations as much as possible, we only considered the spatiotemporal dynamics of bacteria within 50 μm above a square area containing a specific host cell monolayer. Subsequently, we considered the three-dimensional movement of bacteria in the cubic space under the action of Stokes force, Brown force, gravity and buoyancy, where the gravity and buoyancy played important roles in regulating vertical motility of bacteria, adhesion, and remobilization after detachment from the underlying substrates. Generally, the gravity term in equation (2) in the Supplementary Information could not be omitted in the simulation analysis because of the presence of buoyancy around the bacteria. Otherwise, it would be difficult for these simulations to converge to the desired results. These have been further emphasized in the newly revised manuscript (See Page 16, Line 12-18 in the Supplementary Information)

3. *What condition is satisfied and with what probability when bacterium receptor-ligand binds and dissociates?*

Response: Thank you for your valuable comments. In the newly revised manuscript, we have already added the conditions and probabilities of receptor-ligand binding and dissociation. In the simulations, we hypothesized that the density of collagen IV proteins on the micropatterned host cell monolayers was directly related to bacteria-cell adhesion forces and adhesion energies, which were beforehand quantified by FluidFM-based SCFS experiments. Subsequently, we introduced a mechanochemical coupling model to quantitatively characterize the dynamics of ligand-receptor binding/dissociation between a bacterium and the corresponding host cell in our Monte Carlo simulations (See Page 17, Line 34-41 in the Supplementary Information)

References

- 1 Guimaraes, C. F., Gasperini, L., Marques, A. P. & Reis, R. L. The stiffness of living tissues and its implications for tissue engineering. *Nat. Rev. Mater.* **5**, 351-370 (2020).
- 2 Handorf, A. M., Zhou, Y. X., Halanski, M. A. & Li, W. J. Tissue stiffness dictates development, homeostasis, and disease progression. *Organogenesis* **11**, 1-15 (2015).
- 3 He, S. J. *et al.* Stiffness restricts the stemness of the intestinal stem cells and skews their differentiation toward goblet cells. *Gastroenterology* **164**, 1137-1151 (2023).
- 4 Tumsatan, P., Uscharapong, M., Srinakaran, J., Nanagara, R. & Khunkitti, W. Role of shear wave elastography ultrasound in patients with systemic sclerosis. *J Ultrasound* **25**, 635-643 (2022).
- 5 Liu, X. *et al.* Extracellular matrix stiffness modulates host-bacteria interactions and antibiotic therapy of bacterial internalization. *Biomaterials* **277**, 121098 (2021).
- 6 Bastounis, E. E., Yeh, Y. T. & Theriot, J. A. Matrix stiffness modulates infection of endothelial cells by *Listeria monocytogenes* via expression of cell surface vimentin. *Mol. Biol. Cell* **29**, 1571-1589 (2018).

REVIEWERS' COMMENTS

Reviewer #3 (Remarks to the Author):

The authors adequately addressed all my concerns and I have no further comments.

Reviewer #4 (Remarks to the Author):

the points were addressed adequately